



# Observing traveling waves in glaciers with remote sensing: New flexible time series methods and application to Sermeq Kujalleq (Jakobshavn Isbræ), Greenland

Bryan Riel[1], Brent Minchew[1], and Ian Joughin[2]

[1]Department of Earth, Atmospheric and Planetary Sciences, Massachusetts Institute of Technology, Cambridge, MA, USA
[2]Polar Science Center, Applied Physics Lab, University of Washington, Seattle, WA, USA

**Correspondence:** Bryan Riel (briel@mit.edu)

**Abstract.**

The recent influx of remote sensing data provides new opportunities for quantifying spatiotemporal variations in glacier surface velocity and elevation fields. Here, we introduce a flexible time series reconstruction and decomposition technique for forming continuous, time-dependent surface velocity and elevation fields from discontinuous data and partitioning these time series into short- and long-term variations. The time series reconstruction consists of a sparsity-regularized least squares regression for modeling time series as a linear combination of generic basis functions of multiple temporal scales, allowing us to capture complex variations in the data using simple functions. We apply this method to the multitemporal evolution of Sermeq Kujalleq (Jakobshavn Isbræ), Greenland. Using 555 ice velocity maps generated by the Greenland Ice Mapping Project and covering the period 2009 – 2019, we show that the amplification in seasonal velocity variations in 2012 – 2016 was coincident with a longer-term speedup initiating in 2012. Similarly, the reduction in post-2017 seasonal velocity variations was coincident with a longer-term slowdown initiating around 2017. To understand how these perturbations propagate through the glacier, we introduce an approach for quantifying the spatially varying and frequency-dependent phase velocities and attenuation length scales of the resulting traveling waves. We hypothesize that these traveling waves are predominantly kinematic waves based on their long periods, coincident changes in surface velocity and elevation, and connection with variations in the terminus position. This ability to quantify wave propagation enables an entirely new framework for studying glacier dynamics using remote sensing data.

## 1 Introduction

Until recently, observations of glacier and ice stream motion were limited to velocity snapshots measuring motion over distinct time periods, most commonly averaged over multiple years or annually repeating (Rignot et al., 2011; Gardner et al., 2018; Moon et al., 2012). While the increase in spatial coverage of velocity measurements facilitated by the increasing availability of satellite-based remote sensing observations has allowed for ice sheet-wide analysis, the complexity of glacier dynamics requires observations at multiple temporal scales. Rapid responses in ice velocity to changes in external forces, such as ocean melt rate or calving frequency, may be superimposed on longer-term responses to variations in surface melt, ice geometry,





and other factors (Howat et al., 2010; Joughin et al., 2014; Felikson et al., 2017; Wood et al., 2018). Therefore, velocity
observations averaged over multiple years may not resolve rapid dynamical changes, whereas isolated snapshots acquired over
a short time window may bias estimates of longer-term or periodic trends (Minchew et al., 2017). Since the relevant timescales
for resolving glacier dynamics varies significantly from glacier to glacier, any attempt to reconstruct the velocity history must
be able to resolve these multiple temporal scales with minimal prior information.

For the past few decades, continental scale observations of ice motion have been derived from the complementary use of
spaceborne optical imagery and synthetic aperture radar (SAR) data (Scambos et al., 1992; Goldstein et al., 1993; Joughin et al.,
1998; Rignot et al., 2011; Gardner et al., 2018; Joughin et al., 2018). By comparing optical images acquired at different times
over a common area, surface deformation can be quantified using feature-tracking-based techniques (Luckman and Murray,
2005; Dehecq et al., 2015; Fahnestock et al., 2016; Kääb et al., 2016). Optical data from missions such as Landsat 7 and 8,
which have provided optical data for countless studies of surface deformation, have recently been supplemented with data
from Earth-observing missions like Sentinel-2, as well as modern cubesat constellations (Kääb et al., 2017). While optical data
depend on daylight conditions and cloud-free weather, SAR data are able to observe Earth's surface under any condition, thus
allowing for temporally dense coverage over many glaciers and ice streams (Rignot, 1996; Rignot and Kanagaratnam, 2006;
Joughin et al., 2012; Lemos et al., 2018). The last decade has seen the launch of multiple SAR satellites which has led to the
formation of an international constellation of all-weather Earth-observing platforms that can provide unprecedented spatial
and temporal resolution over many areas of interest. At the same time, several researchers have synthesized these multiple data
sources to consistently produce repeating ice velocity products over Greenland, Antarctica, and other dynamic areas of the
cryosphere. (Joughin et al., 2010, 2011; Nagler et al., 2015; Mouginot et al., 2017; Gardner et al., 2019). These products, many
of which are publicly available, have simplified access to high-quality velocity observations, allowing for a new era of rapid
assessment and quantification of ice motion over the most critical regions.

In this work, we utilize velocity products generated by the Greenland Ice Mapping Project (GIMP), which has used data
from a variety of satellites and sensors to observe ice sheet change over Greenland since 2000 (Joughin et al., 2018). In
particular, we will focus on forming a temporally continuous time-dependent velocity dataset over Sermeq Kujalleq (hereafter
referred to as Jakobshavn Isbræ) using high spatial resolution velocity data generated with the German Aerospace Center's
(DLR) TerraSAR-X mission (Joughin et al., 2020). We present a flexible time series decomposition method that allows us to
isolate short- and long-term variations in the velocity data while also allowing for sub-epoch quantification of velocity changes
throughout the glacier. This method, coupled with the 11-day repeat time for the TerraSAR-X velocities from 2009–2019,
allows us to investigate numerous changes to the flow characteristics of Jakobshavn Isbræ over the past decade. For example,
the seasonal variations in velocity magnitude that became more prominent following the disintegration of the floating ice
tongue in 2004 experienced further amplification in 2012 (Joughin et al., 2012). Coincident with the seasonal amplification
was an increase in the average ice velocity from 2012 to 2016. Both of these signals have been hypothesized to be driven
primarily by changes in the position of the terminus (Joughin et al., 2012; Bondzio et al., 2017). Starting in winter 2016,
this trend reversed: average ice velocities decreased over the course of three years while the seasonal variations decreased in
amplitude (Joughin et al., 2018; Khazendar et al., 2019; Joughin et al., 2020). Thus, the complex velocity history at Jakobshavn





Isbræ over the past decade provides a unique test case for assessing the quality and feasibility of the time series decomposition
method presented here. Specifically, the repeated terminus-driven velocity perturbations at multiple timescales admits a new
framework for investigating the mechanics of glaciers and ice streams.

## 2  Time Series Analysis Methods

Geodetic time series contain measurements of geophysical processes with variable spatial and temporal scales. Over glaciers,
mesoscale changes in precipitation or climate may induce slow transient and widespread changes in ice-surface elevation,
while calving events at glacier termini and thinning of ice shelves can generate traveling waves that propagate upstream over a
wide range of timescales (Hewitt and Fowler, 2008; Fowler, 2011; Minchew et al., 2017). Many external forcing functions can
result in non-linear variations in internal ice dynamics due to factors like the non-Newtonian viscosity of ice, softening of ice
in shear-margins by viscous dissipation, lubrication of glacier beds due to surface melt, and changes in gravitational driving
stress taking effect (Schoof, 2010; Minchew et al., 2018a; Meyer and Minchew, 2018). The effects of these processes are often
additive and collocated, so measurements of ice surface velocity and elevation with sufficient temporal sampling will record
the combined effect of all processes. Isolating the spatial and temporal signature of each distinct geophysical mechanism is
necessary for identifying the appropriate forcing function and inferring physical properties of the glacier.

In this study, we generalize previous surface velocity time series methods (Minchew et al., 2017), which were restricted to
sinusoidal variations in time, by modeling temporal variations in surface velocity as a linear combination of reference functions
that resemble typical signals observed in geodetic time series (Hetland et al., 2012; Riel et al., 2014, 2018). These reference
functions can be non-orthogonal and are placed in a large *dictionary* (matrix), $\mathbf{G} \in \mathbb{R}^{M \times N}$, such that the temporal model for a
time series at a given location is linear and given as

$$\mathbf{d} = \mathbf{G}\mathbf{m} + \mathcal{N}(\mathbf{0}, \mathbf{C_d}), \tag{1}$$

where $\mathbf{d} \in \mathbb{R}^{M \times 1}$ is the vector of observations, $\mathbf{m} \in \mathbb{R}^{N \times 1}$ is the coefficient vector solution, and $\mathbf{C_d} \in \mathbb{R}^{M \times M}$ is the covariance
matrix corresponding to zero-mean Gaussian observation errors $\mathcal{N}(\mathbf{0}, \mathbf{C_d})$. An important advantage of using a linear model to
represent the time series is the ability to evaluate the model at any arbitrary time, which provides a natural way to assimilate
time series with missing or irregularly spaced data. Additionally, linear models facilitate the use of powerful and efficient linear
regression inverse methods to solve for the coefficients in $\mathbf{m}$ (Tarantola, 2005).

The dictionary $\mathbf{G}$ can contain any combination of functions that collectively capture the observable temporal variations.
Thus, the inverse problem for $\mathbf{m}$ is often ill-posed because the dictionary $\mathbf{G}$ can be overcomplete, with many more refer-
ence functions (columns) than observations (rows). Therefore, we use regularized least squares to obtain an estimate $\hat{\mathbf{m}}$ that
minimizes a cost function containing the data residual and regularization terms, such that (Riel et al., 2014, 2018)

$$\hat{\mathbf{m}} = \underset{\mathbf{m}}{\mathrm{argmin}} \left\{ \|\mathbf{d} - \mathbf{G}\mathbf{m}\|^2_{\mathbf{C_d}} + \mathbf{m}^T \mathbf{C_m}^{-1} \mathbf{m} + \lambda \|\mathbf{m}\|_1 \right\}, \tag{2}$$

where $\| \cdot \|_{\mathbf{C_d}}$ denotes the Euclidean or $\ell_2$-norm that accounts for noise in the observations via the data covariance matrix $\mathbf{C_d}$,
$\mathbf{C_m} \in \mathbb{R}^{M \times M}$ is a prior covariance matrix that represents expected statistics of the coefficients (i.e., *a priori* information),





and the final term $\lambda\|\mathbf{m}\|_1$ is an $\ell_1$-norm term that encourages a sparse number of non-zero coefficients. The function in curly brackets in Eq. 2 is a convex cost function, which provides a solution that is guaranteed to be globally optimal. Implementation and the procedure for solving Eq. 2 for $\hat{\mathbf{m}}$ is detailed by Riel et al. (2014).

The coefficient $\lambda$ in Eq. 2 is a penalty parameter controlling the strength of the sparsity-inducing regularization. Schemes
for choosing values of $\lambda$ based on the amount of data available and the desired smoothness of the solution are discussed by Riel et al. (2014). In practice, the $\ell_1$-norm regularization can be applied to a subset of $\mathbf{m}$, which is assumed to be sparse, and depending on the reference functions in the dictionary that correspond to this subset, fewer non-zero coefficients may result in a smoother time series reconstruction. This regularization approach results in a compact representation for transient variations, which can aid in determining the dominant timescales and onset times captured in the data while potentially improving
detection of signals with a lower signal-to-noise ratio (SNR).

In this study, we model transient signals as linear combinations of third-order integrated B-splines ($\mathrm{B}^i$-splines), which exhibit one-sided behavior of a particular timescale (Hetland et al., 2012; Riel et al., 2014). By allowing for combinations of $\mathrm{B}^i$-splines of different timescales and onset times, we can reconstruct a wide variety of transient signals. Strong seasonal variations in ice surface velocity and elevation, like those observed on Jakobshavn Isbræ, exhibit large changes in amplitude from year-to-year
(Joughin et al., 2010, 2018). Therefore, we use a linear combination of third-order B-splines to reconstruct seasonal signals with time-varying amplitudes (Riel et al., 2018). To encourage seasonal coherency of the B-spline coefficients, we construct $\mathbf{C_m}$ in Eq. 2 such that the B-splines co-vary with other B-splines that share the same centroid time within any given year. The covariance strengths are constructed to decay exponentially in time. The flexibility in representing potentially complex temporal variations afforded by this approach avoids the severe limitations of using a single or small subset of sinusoidal variations (e.g.,
Minchew et al., 2017) and allows for a framework of transient and periodic variations that readily admit physical interpretation. The interpretability of the resulting posterior model $\hat{\mathbf{m}}$ (which in this study primarily represents surface flow speeds) in terms of external drivers and intrinsic dynamics of the glacier is a marked advantage of our approach over time series approaches based on singular value decomposition (e.g., Samsonov, 2019). This advantage is amplified when using the sparse regularization techniques to constrain the timing, duration, and amplitude of transient events that are superimposed on periodic variations, as
we describe in this study. Importantly, the framework described by Eq. 2 also enables quantification of the formal uncertainty estimates in the inferred time series (i.e., posterior model $\hat{\mathbf{m}}$).

Uncertainties for the estimated model coefficients can be formally quantified by combining observational uncertainties contained in the data covariance matrix $\mathbf{C_d}$ with the dictionary $\mathbf{G}$ and prior covariance matrix $\mathbf{C_m}$ (Tarantola, 2005; Bishop, 2006):

$$\tilde{\mathbf{C}}_\mathbf{m} = \left(\mathbf{G}^T\mathbf{C_d}^{-1}\mathbf{G} + \mathbf{C_m}^{-1}\right)^{-1}, \tag{3}$$

where $\tilde{\mathbf{C}}_\mathbf{m}$ is the posterior model covariance matrix. Lower coefficient uncertainties can thus be obtained by a combination of reduced data noise and lower prior uncertainties on those coefficients. Similarly, the posterior covariance matrix of the reconstructed time series can be formally computed as (Tarantola, 2005; Bishop, 2006):

$$\tilde{\mathbf{C}}_\mathbf{d} = \mathbf{G}\tilde{\mathbf{C}}_\mathbf{m}\mathbf{G}^T. \tag{4}$$





In general, the structure of $\mathbf{G}$, in particular the non-orthogonality of the included reference functions, will have a strong effect on the covariances between different model coefficients and time epochs via the off-diagonal values in the matrices $\tilde{\mathbf{C}}_{\mathbf{m}}$ and $\tilde{\mathbf{C}}_{\mathbf{d}}$, respectively. Properly quantifying these uncertainties and covariances is critical for any subsequent interpretation or analyses using the modeled time series.

After using Eq. 2 to estimate $\hat{\mathbf{m}}$ for a given data time series $\mathbf{d}$, the seasonal and transient signals can be reconstructed as

$$\hat{\mathbf{d}}_S = \sum_{j \in S_S} \mathbf{G}_j \mathbf{m}_j,$$
$$\hat{\mathbf{d}}_T = \sum_{j \in S_T} \mathbf{G}_j \mathbf{m}_j, \tag{5}$$

where $\hat{\mathbf{d}}_S$ and $\hat{\mathbf{d}}_T$ are the reconstructed seasonal and transient signals, respectively, $S_S$ represents the set of indices corresponding to the seasonal B-splines, $S_T$ represents the set of indices corresponding to the transient $\mathbf{B}^i$-splines, and $\mathbf{G}_j$ is the vector corresponding to the $j$-th column of $\mathbf{G}$. Correspondingly, we can use these reconstructed signals to *detrend* the data, e.g.

$$\mathbf{d}_S = \mathbf{d} - \hat{\mathbf{d}}_T,$$
$$\mathbf{d}_T = \mathbf{d} - \hat{\mathbf{d}}_S. \tag{6}$$

Examples of the practical implementation of these multitemporal methods are provided in the Results section.

In some cases it may be desirable to enforce spatial coherency when solving for $\hat{\mathbf{m}}$ to reduce the influence of data noise on $\hat{\mathbf{m}}$ (Hetland et al., 2012; Riel et al., 2014; Minchew et al., 2015). Examples of such cases involve low-amplitude signals with spatial wavelengths longer than a single data pixel and, more generally, cases where the data have low SNR values. The
software made available with this study allows the user to enforce spatial coherency (cf. Riel et al., 2018), but for the velocity and elevation time-dependent data sets used in this study, we do not enforce spatial coherence between neighboring pixels because doing so can be computationally expensive and is unnecessary in our case. Instead, we solve for $\hat{\mathbf{m}}$ for each pixel independently and justify this decision based on the fact that the SNR is generally high over Jakobshavn Isbræ, reducing the
need for spatial coherency in the inversion process.

## 3 Data

Data used in this study provide information on the time-dependent surface velocity and elevation fields. In both cases, the raw remote sensing data were processed to individual time-stamped fields and made publicly available through different projects and publications. In this section, we briefly describe these data sets.

### 3.1 Surface Velocity Data

The Greenland Ice Mapping Project (GIMP) produces comprehensive horizontal ice-surface velocity time series for the Greenland Ice Sheet using a variety of satellites and sensors (Joughin et al., 2010, 2018, 2020). These data allow for widespread





observations of glacier velocity variations with increasing temporal resolution as more data from more sensors became available. Over select glaciers like Jakobshavn Isbræ, 11-day and monthly repeat TerraSAR-X/TanDEM-X (TSX) SAR pairs for
both ascending and descending orbits are available and provide much higher temporal resolution than is available on many other glaciers. The GIMP velocity maps are formed using speckle-tracking techniques on each SAR pair (Joughin, 2002). Speckle tracking is generally more robust to large-scale displacements than standard interferometric methods and allows for uncertainty quantification through estimates of the statistics of correlation measurements. These formal errors are provided with the GIMP velocity fields.

In this work, we use 555 GIMP horizontal velocity fields generated from TSX data covering the period 2009 – 2019. The velocity fields are provided with 100-m grid spacing, and the short repeat times allow for mostly complete spatial coverage of Jakobshavn Isbræ due to the high number of coherent surface features facilitating high SNR for the speckle tracking techniques. During the course of this work, we discovered systematic discrepancies in velocity measurements between SAR pairs collected from ascending and descending orbits at the same geographic location and approximately the same time. These discrepancies
are generally limited to the shear margins of the glacier and cluster in areas much smaller than the glacier. To improve the overall quality of the time series, we masked out areas where the velocities collected from near-coincident ascending and descending orbits differed by more than 250 m/yr.

To extend the velocity time series, we include monthly GIMP velocity maps derived from optical Landsat imagery for the years 2004 – 2009 (Jeong et al., 2017). This longer time series allows us to compare long-term trends in surface velocity to
long-term changes in glacier terminus position. The optical velocity maps are provided at 100-m grid spacing on a grid that is not aligned with the TSX velocity maps. Thus, we resampled the Landsat-derived velocity fields to the same grid as the TSX velocity fields.

## 3.2 Surface Elevation Data

Ice surface elevation will rise and fall in response to variations in mass balance and ice flow. While time-dependent measure-
ments of surface elevation are generally available at a lower temporal resolution than measurements of velocity, the increased availability of optical imaging satellites and robustness of photogrammetry techniques has allowed for the generation of time-dependent digital elevation model (DEM) strips at sub-annual epochs. Here, we use publicly available DEMs from ArcticDEM and the Oceans Melting Greenland (OMG) mission.

The ArcticDEM initiative automatically produces 2-meter resolution DEM strips over all land area north of 60° latitude using
stereo auto-correlation techniques (Porter et al., 2018). We use DEMs generated from optical data collected over Jakobshavn from 2010 to 2017. The geographical extent of the strips and their associated acquisition times are irregular, so we interpolate the strips onto a uniform spatial grid using inverse distance weighting while using nearest neighbor interpolation in time to sample the strips onto a uniform temporal grid.

Because ArcticDEM data are available from 2010 to 2017, a timescale shorter than the GIMP surface velocity data, we
augment the surface elevation time series with DEMs produced annually by the Oceans Melting Greenland (OMG) mission in March 2017, 2018, and 2019. The OMG DEMs were constructed with Ka-Band single pass SAR interferometry from the





Glacier and Ice Surface Topography Interferometer (GLISTIN-A) instrument, providing high-precision (< 50 cm) elevation maps over 10-12 km-wide swaths over various marine-terminating glaciers in Greenland (OMG Mission, 2016). These interferometric products were processed to an intrinsic resolution of 3 meters and then georegistered to a ground spacing of approximately 3 meters. For comparison with velocity data, both ArcticDEM and OMG DEMs were resampled to a grid spacing of 100 meters.

### 3.3  Calving front positions

To elucidate the connection between observed changes in the terminus position and observed variations in ice surface velocity and elevation, we use calving front positions from a variety of sources. The main source are calving front positions automatically determined from TSX images acquired from 2009 to 2015 (Zhang et al., 2019). We supplement these data with calving front locations provided by the Greenland Ice Sheet Climate Change Initiative (CCI) from 2009 to 2016, which were computed using manual digitization of ERS, Sentinel-1, and LANDSAT imagery (ESA, 2016). We further supplement the data with our own manual digitization of the calving front from TerraSAR-X images, skipping months where the front position is not clear (Joughin et al., 2020). The upper bound of horizontal position errors for these data are approximately 200 meters, which indicates sufficient accuracy for the qualitative comparison with ice velocities performed here.

To develop a time series of calving front positions for comparison with our velocity fields, we first choose the reference position for the calving front to be a feature in the bedrock topography with reported dynamical implications for flow variations on Jakobshavn. Cassotto et al. (2019) suggested that the acceleration in velocities in 2012 corresponded to a retreat of the calving front passed a narrow, shallow portion of the bed topography, which acts as a pinning point that facilitates higher extensional and lateral shear stresses relative to the wider and deeper basin upstream (Morlighem et al., 2017). This suggestion is a generalization of the often-mentioned process of retreat of the calving front into an overdeepened basin (e.g. Nick et al., 2009; Joughin et al., 2012) and has been reproduced in three-dimensional models incorporating detailed bed topography (e.g. Morlighem et al., 2016). We therefore generate a time series of the intersection of the calving front with the glacier centerline, referenced to the downstream position of the aforementioned shallow bed location. To reduce time series noise and facilitate comparison with the velocity time series, we again apply our linear least squares method to represent the front time series as a combination of B-splines for short-term, seasonal signals and integrated B-splines for long-term transient signals. Since the absolute position of the front relative to the shallow bed location is important, we do not isolate the front seasonal (short-term) signal from the time series in the subsequent analysis.

## 4  Results

Hereafter, we focus on applying the time series analysis methods presented in Section 2 to analyze and decompose the observed time-dependent velocity magnitude (speed) and surface elevation fields summarized in Section 3 into sub-annual (primarily seasonal) and multi-annual transient variations. Our focus is on quantifying the rates and distances over which stress perturbations of various frequencies propagate through Jakobshavn. We use only the time-dependent flow speeds because Jakobshavn




flows along a deep, narrow channel in the underlying bed, which leads to temporally consistent velocity unit vectors throughout

the observation period. Future work will be aimed at adding multidimensional capability to the time series analysis methods we introduce in this study. Before proceeding to the analysis of the time-dependent velocity fields, we note for completeness that we observe magnitudes and spatial patterns of mean flow speed that are consistent with previous studies, with the highest speeds near the terminus decaying quasi-exponentially with upstream distance (Figure 1A,B; Movie S1).

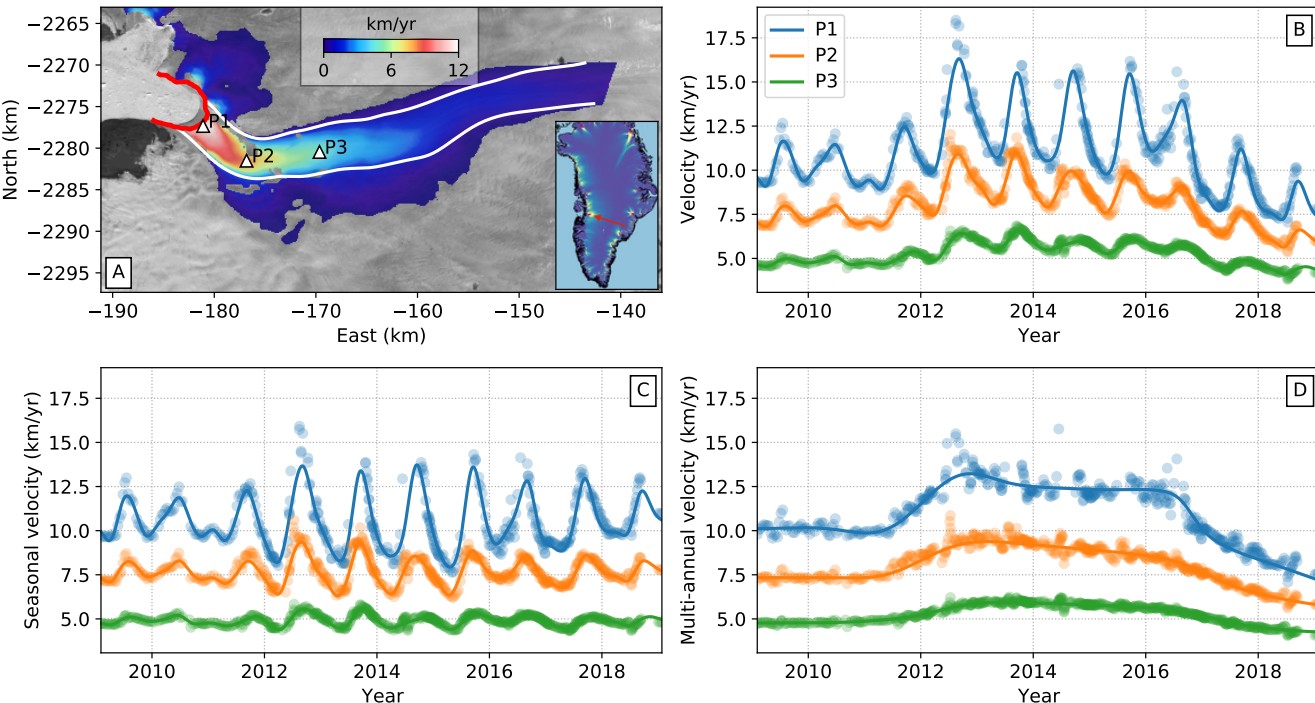

**Figure 1.** Sermeq Kujalleq (Jakobshavn Isbræ) mean velocity field and select velocity time series. A) Mean velocity between 2009 and 2019. The background image is a Landsat 8 image acquired on August 2017. White lines correspond to approximate glacier boundaries as defined by peak shear strain rate, and the red line indicates the winter 2017 terminus position. White triangles indicate the points P1, P2, and P3 from which data shown in panels B–D are taken. Inset shows (with red arrow) the approximate study area within Greenland with mean velocities from Joughin et al. (2011). B) Time-dependent speed at points P1, P2, and P3. C) Short-term velocity time series showing predominantly seasonal variations. Mean velocities are added to time series for visual clarity. D) Long-term velocity time series showing the 2012 speedup and 2017 slowdown. In panels B–D, solid lines show our model results while dots indicate (B) observed speeds or (C and D) detrended observations (see Eq. 6).

## 4.1 Seasonal Variations in Surface Velocity

The reconstructed velocity time series demonstrates the ability of our flexible method to smoothly interpolate the velocity data in time in a manner that preserves the seasonal variations. In particular, the use of temporally coherent B-splines to model





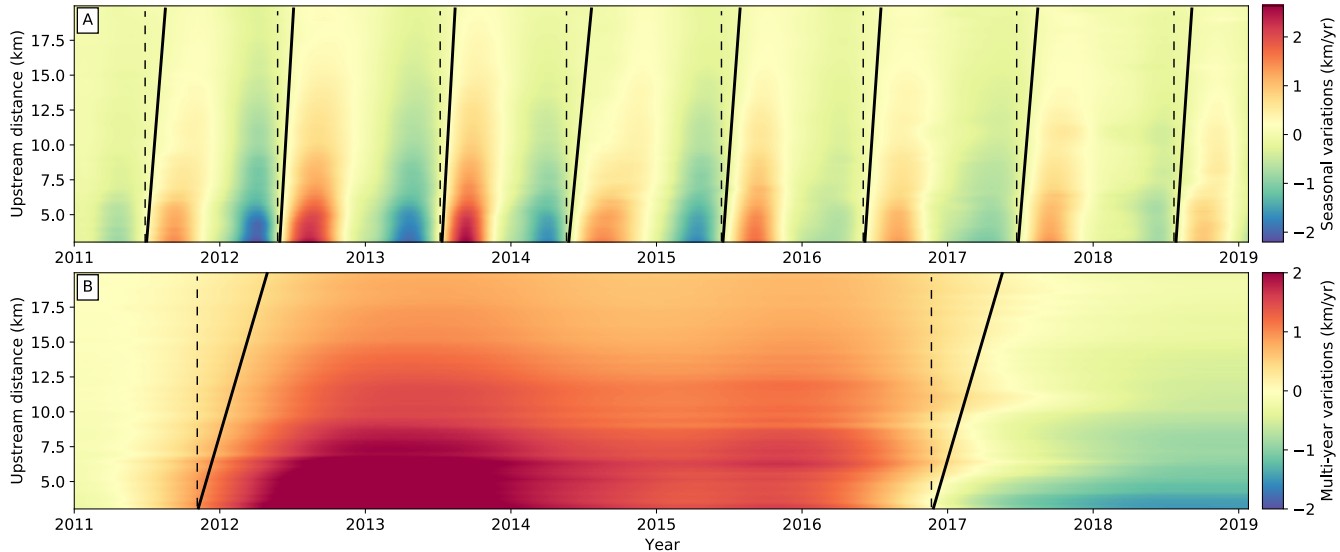

**Figure 2.** Temporal evolution of glacier flow variations along a centerline segment for decomposed seasonal (A) and multi-year (B) signals. The centerline trace is shown in map view in Figure 4, and distance values here are measured upstream along the centerline from the winter 2017 terminus position. Solid black lines approximately correspond to the leading edges of propagating positive velocity variations. In panel B, the leading edge for the 2017 slowdown is also shown. Vertical black dashed lines indicate onset times for the propagating wave initiating at the terminus and are equivalent to the propagation path for a wave with infinite propagation speed. The tangent of the angle between solid and dashed lines is the phase velocity averaged over the observable propagation distance. The marked difference in phase velocities between the seasonal and multi-year signals indicates (frequency) dispersion.

seasonal variations allows for reconstruction of several summer speedup events where data happen to be more sparse for certain years (Figure 1). By applying Eq. 5 to decompose the velocity magnitude time series into seasonal and transient components, we show that short-term velocity variations on Jakobshavn are dominated by the seasonal cycle of summer speedup and winter

slowdown. In this section, we focus on these seasonal variations, leaving discussion of the multi-annual transient variations for the next section.

The flexibility of the B-spline representation for the seasonal time series allows us to quantify the change in the amplitude of summer speedup from year-to-year and at each point on the glacier. In Figures 1 and 2 we show these variations in two different views to aid in interpretation of the results. Figure 1A,B represents a classical view of spatiotemporal variations

in surface velocity, with a map of secular velocity (Figure 1A) and time series of select points on the glacier (Figure 1B). Figure 1C shows, for the same points on the glacier, the seasonal variations, which are the total signal shown in Figure 1B less the inferred multi-year trends discussed in the next section. In Figure 2, we present a space-time plot for the (A) seasonal and (B) multi-year variations along the centerline transect shown in Figure 4A. This representation allows for an intuitive visualization of spatiotemporal variations in the surface velocity fields and, most relevant for this study, the propagation of





velocity variations through the glacier in time. Our analysis focuses on this propagation by treating velocity variations as traveling waves with quantifiable attenuation and propagation rates. To aid in this discussion, we have provided a visual representation of the upstream propagation rate in Figure 2 using the solid and dashed black lines. The angle between the two lines represents the phase velocity $c_p$, an important concept in this study that is defined as the speed at which the phase of a wave of a given frequency $\omega$ travels. Thus, $c_p = \omega/k_r$ where $k_r$ is the real component of the angular wavenumber.

The results shown in Figures 1C and 2A indicate that the amplitudes of seasonal velocity variations are largest near the terminus and decay as a function of upstream distance. By extracting a centerline transect of amplitude (averaged over all observed seasons) as a function of distance, we estimate an attenuation (or $e$-folding) length scale of approximately $7 \pm 0.3$ km for all observed seasonal variations (Figure 3A), which implies that large-amplitude velocity variations near the terminus position are observable at farther upstream distances relative to smaller amplitude variations. This effect can be observed by

comparing in map view seasonal velocity variations for years where the amplitudes are markedly different (Figure 4A–B). For the years 2009 – 2011 and 2017 – 2018, peak amplitudes of seasonal velocity variations did not exceed 3 km/year, whereas for the years 2012 – 2015 (the period with the fastest glacier flow speeds in our observations) the highest amplitudes exceed 6 km/year. Thus, there is a clear correlation between mean flow speed for a given year and the amplitude of seasonal variations, which we will explore in later sections.

In addition to attenuation, we are interested in constraining the rate of propagation of surface velocity variations. We quantify these variations in terms of phase velocities by constraining the relative timing of peak velocity for different variation temporal frequencies. As with amplitude, we present the absolute and relative timing of the velocity variations in multiple ways to help build a more intuitive framework for the reader. This presentation follows the same structure as the amplitude variations discussed in detail above, with a classical view shown in Figure 1, the space-time diagram in Figure 2, the mean over all

seasons along a centerline transect in Figure 3B, and map view of the relative timing in Figure 4C (with formal uncertainties in timing given in Figure 4D).

For seasonal variations in surface velocity, the time of peak velocity varies slightly from year-to-year, with the earliest peaks occurring around mid-August and the latest peaks occurring around mid-September. However, the spatial pattern of relative timing in the upstream direction from the terminus is broadly consistent even among years with large differences in mean

velocities, which indicates a common mechanism for the seasonal cycle (Figure 2A). As expected for marine-terminating glaciers like Jakobshavn, the timing of the peak seasonal signal indicates that seasonal variations originate at the terminus and propagate upstream (Figures 1C and 2A).

To investigate the spatial characteristics of the timing of peak velocity, we fit a simple temporal model using a sum of sinusoids to the velocity data from 2011 to 2018 at each pixel with the estimated long-term signals removed, $\mathbf{d}_S$ (e.g., Figure

1C) such that

$$\mathbf{d}_S = C_0 + \sum_i \left[ C_i \cos\left(\omega_i \mathbf{t}\right) + S_i \sin\left(\omega_i \mathbf{t}\right) \right], \tag{7}$$

where $\omega_i$ is the angular frequency for the $i$-th sinusoid, $C_i$ and $S_i$ are the coefficients of the cosine and sine components, respectively, and $C_0$ is a constant offset. After estimating the values of $C_i$ and $S_i$, the amplitude and phase (*i.e.*, relative timing)

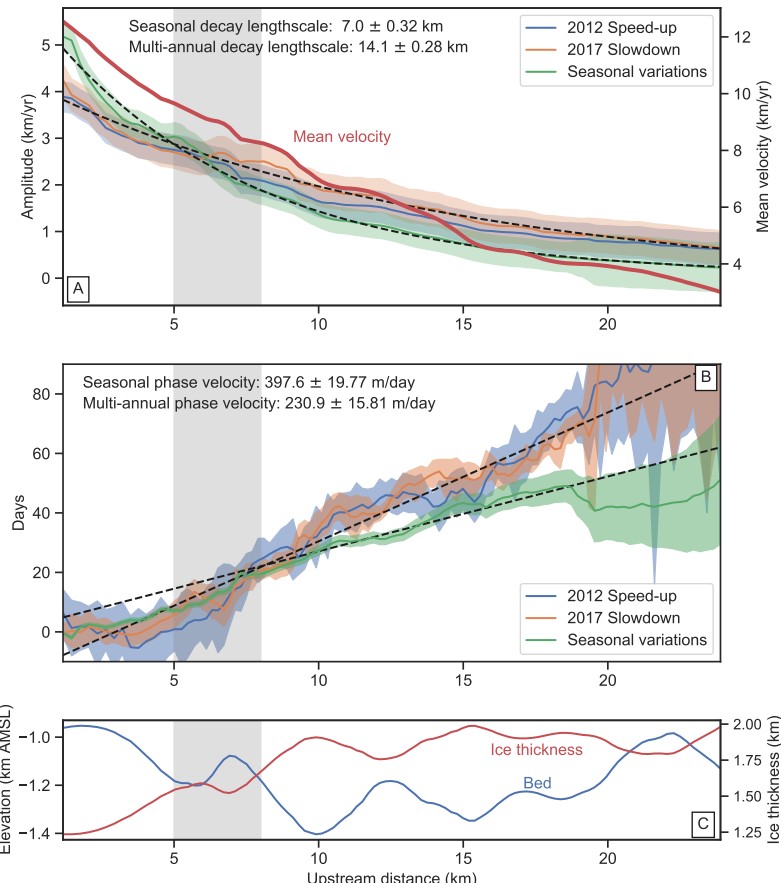

**Figure 3.** Glacier centerline transects of phase delay and velocity variation amplitude for the seasonal, 2012 speedup, and 2017 slowdown signals compared to centerline ice thickness and bed depth. A) Velocity variation amplitude and 1-standard-deviation uncertainties for the three different signals. The decay lengthscale (*e*-folding distance) for the multi-annual signals is roughly twice that of the seasonal signal (represented by best-fit exponential decay model in black dashed lines). Dark red line corresponds to the mean velocity magnitude along the centerline. B) Relative phase delay for the three different signals: seasonal phase delay - green; 2012 speedup - blue; 2017 slowdown - orange. The upstream centerline distance is with respect to the intersection of the centerline and the 2017 terminus. For distances greater than 8 km upstream, the phase velocity for the seasonal signal is roughly twice as fast as the phase velocities for the multi-annual transients (represented by black dashed lines). The shaded areas represent the 1-standard-deviation formal uncertainties for the annual phase (green) and bootstrapped standard deviation for the transients (blue and orange). C) Centerline ice thickness (red) and bed depth (blue) using bed data from BedMachine V3 (Morlighem et al., 2017). For all plots, the gray shaded region represents the upstream region encompassed by the southern bend.

of each sinusoid can be recovered as

$$a_i = \sqrt{C_i^2 + S_i^2},$$ (8)

$$\phi_i = \tan^{-1}(C_i/S_i).$$ (9)



While this model cannot accurately reproduce the amplitude changes or nonsinudoidal variations (e.g. Joughin et al., 2008, 2014), the seasonal phase can be estimated robustly with seven years of data. Furthermore, we can compute the formal phase uncertainties following the procedure outlined in Minchew et al. (2017). The estimated seasonal phase is thus equivalent to
the mean time of peak seasonal velocity for the 2011 – 2018 period while the phase uncertainty is proportional to the formal variance of the mean.

The seasonal phase map shows upstream transmission of velocity perturbations originating at the terminus (Figures 3B and 4C). This propagation occurs rapidly in the first 5 km upstream of the terminus and then slows to a near-constant phase velocity of $398 \pm 20$ m/day (approximately $146 \pm 7$ km/yr), which is more than an order of magnitude faster than the mean flow speed
near the terminus. Our estimates of the phase velocity within the first 5 km upstream of the terminus are limited by the temporal resolution of the data, but by considering the uncertainties in timing, we estimate that the phase velocity in this region must be at least 500 m/day (182.5 km/yr), or approximately 18 times the local mean glacier flow speed. In the across-flow direction, about 8 km upstream from the 2017 calving front, the center of the glacier reaches its peak velocity earlier than the margins by about 15 days, indicating a nonlinear relationship between time-dependent lateral shear strain rates and centerline velocity,
meaning that the effective width of the shear margins (defined here as the centerline velocity divided by the maximum shear strain rate) must change over the seasonal cycle. Finally, we note that the phase uncertainty is generally lowest in the center of the glacier where amplitudes are higher and increases with distance upstream as the amplitudes decay (Figure 4D), as expected from the formal uncertainties (Minchew et al., 2017).

## 4.2   Multi-Year Variations in Surface Velocity

After isolating the long-term signals from the short-term seasonal signals, we can observe clear variations in multi-annual amplitudes at different points along the glacier (Figures 1D and 2B). The temporal density of the velocity time series allows us to quantify spatial variations in the amplitude and timing of the positive and negative multi-annual trends, much like we did with seasonal velocity variations in the previous sections. We observe two events in the data: a speedup that begins in 2012 and a slowdown that begins in late 2016 near the terminus, which we refer to as the 2017 slowdown. We present the results
for multi-annual variations using the same general structure as for the seasonal, with the classical view shown in Figure 1, the space-time diagram in Figure 2, along a centerline transect in Figure 3, and map view of the amplitudes and phase values in Figure 5.

The spatial pattern of the amplitudes of multi-annual velocity variations is remarkably consistent between the two observed events, with the highest amplitudes at the terminus and an exponential decay with distance upstream (Figures 3A and 5A–
B). Notably, the velocity variations induced by these events have an attenuation ($e$-folding) length scale of approximately $14.1 \pm 0.3$ km, which is about twice the attenuation length for seasonal variations (Figure 3A). As a result, we are able to observe multi-annual velocity variations farther upstream than the seasonal timescale velocity variations (Figure 2B).

From the phase delay of the transient signals, we can see that both the 2012 speedup and 2017 slowdown signals originate at the terminus, propagate rapidly along the first 5 km of the glacier, slow down through the southern bend between 5 - 8 km, and
propagate upstream from 8 - 20 km at a generally consistent phase velocity (Figures 3 and 5C–D). Beyond 20 km upstream, the





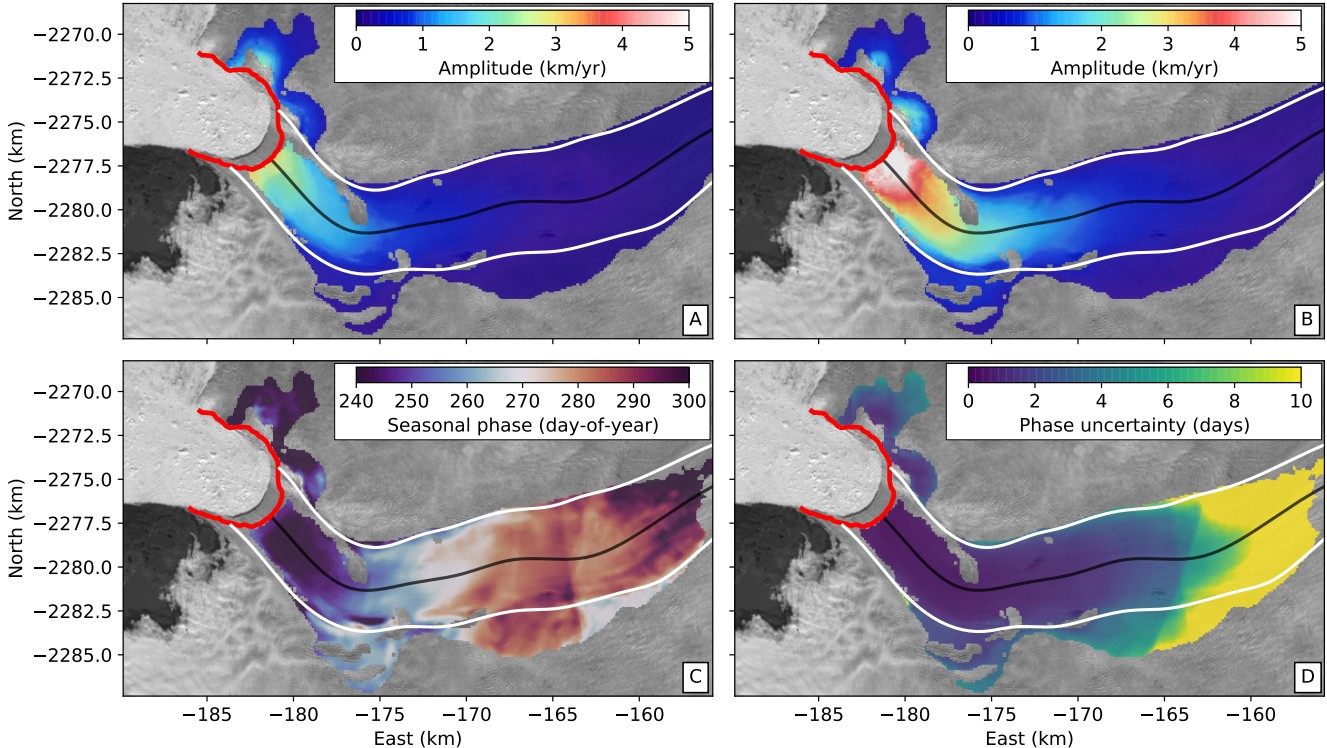

**Figure 4.** Seasonal variations in flow speed and timing of peak velocities. A) Mean seasonal velocity amplitude for the years $2009-2011$ and $2017-2018$ (years not associated with the increased velocities between 2012 and 2015). B) Mean seasonal velocity magnitude variation for the years $2012-2015$. C) Mean day-of-year of peak seasonal velocity (i.e., seasonal phase) for entire observation period. D) Seasonal phase uncertainty ($1\sigma$). Seasonal amplitudes are measured as the difference between the summer high and winter low velocities in the short-term time series as shown in Figure 1C. The highest amplitudes occur at the terminus and decay exponentially upstream.

amplitudes for the velocity variations become too low to reliably estimate the timing of arrival of the transient signals (Figure 5D). The phase velocity between 8 and 20 km upstream is approximately $231\pm16$ m/day ($84\pm6$ km/yr), which is a little more than half of the phase velocity for the seasonal signal and roughly seven times the mean flow speed near the terminus. The upstream phase velocity also has features that suggest a sensitivity to ice thickness and bed topography (Figure 3C), but more

work is needed to establish concrete connections. As with the observed seasonal variations, the phase velocity in the first 5 km upstream of the terminus is at the limit of the temporal resolution of the data with a lower bound on the phase velocity of at least 500 m/day (182.5 km/yr), or approximately 18 times the mean glacier flow speed in this region. While the slowdown in wave propagation in the southern bend is coincident with a local high in the bed topography, more work is needed to evaluate whether the topographic effect is the dominant control on wave propagation. The apparent slowdown may also be an artifact

of numerical errors caused by tracking of peak acceleration/deceleration rather than a multi-year average of sinusoidal phase





as for the seasonal signal. Nevertheless, the consistency in the spatial distribution of peak timing and amplitude reinforces the notion that a common physical mechanism is responsible for multi-year and seasonal velocity variations.

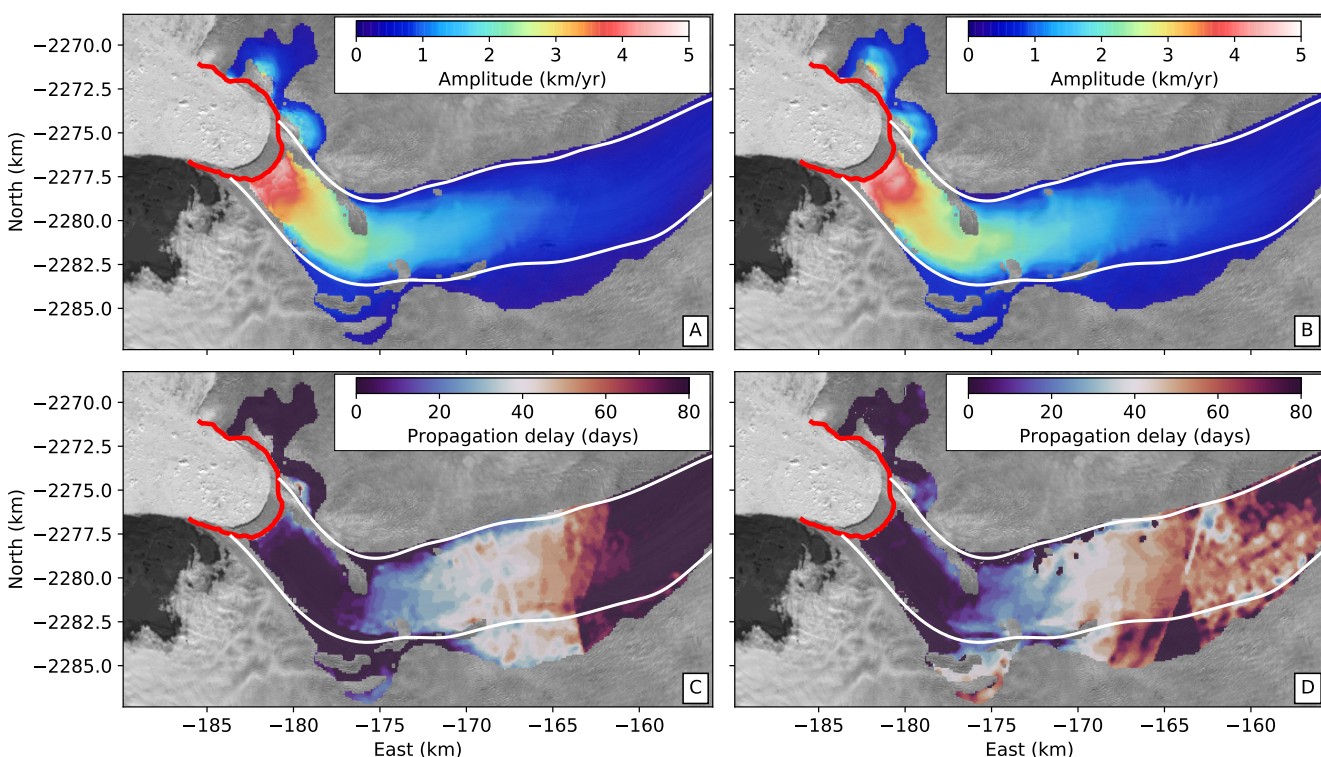

**Figure 5.** Velocity variation amplitude and phase delay maps for 2012 transient speedup (A and C) and 2017 transient slowdown (B and D). In addition to the spatial distribution of phase delay and amplitude being consistent for the two events, these transient phase delays show strong similarities with the seasonal phase delay. However, the transient amplitudes show farther upstream propagation than the seasonal amplitudes.

### 4.3 Multi-Year Variations in Surface Elevation

Ice surface elevation varies in response to changes in snow accumulation and melt (the sum of which constitutes the surface mass balance; SMB), firn compaction (Herron and Langway, 1980; Huss, 2013; Meyer et al., 2019a), and dynamic thinning (thickening) in response to increases (decreases) in the flux divergence of the ice. The interplay between observed elevation and velocity changes at different temporal and spatial scales can thus yield insight into the mechanisms driving longer-term elevation and velocity changes.

For this work, the temporal sampling of the available elevation data (ArcticDEM and OMG DEMs) permits only the comparison of longer-term variations in velocity and elevation. Thus, we compare the long-term velocity and elevation changes for





four separate periods: 1) January 2011 - November 2012; 2) November 2012 - June 2015; 3) January 2016 - June 2017; and 4)
June 2017 - January 2019 (Figure 6). We observe a clear association between the 2012 speedup and lowering of the ice surface

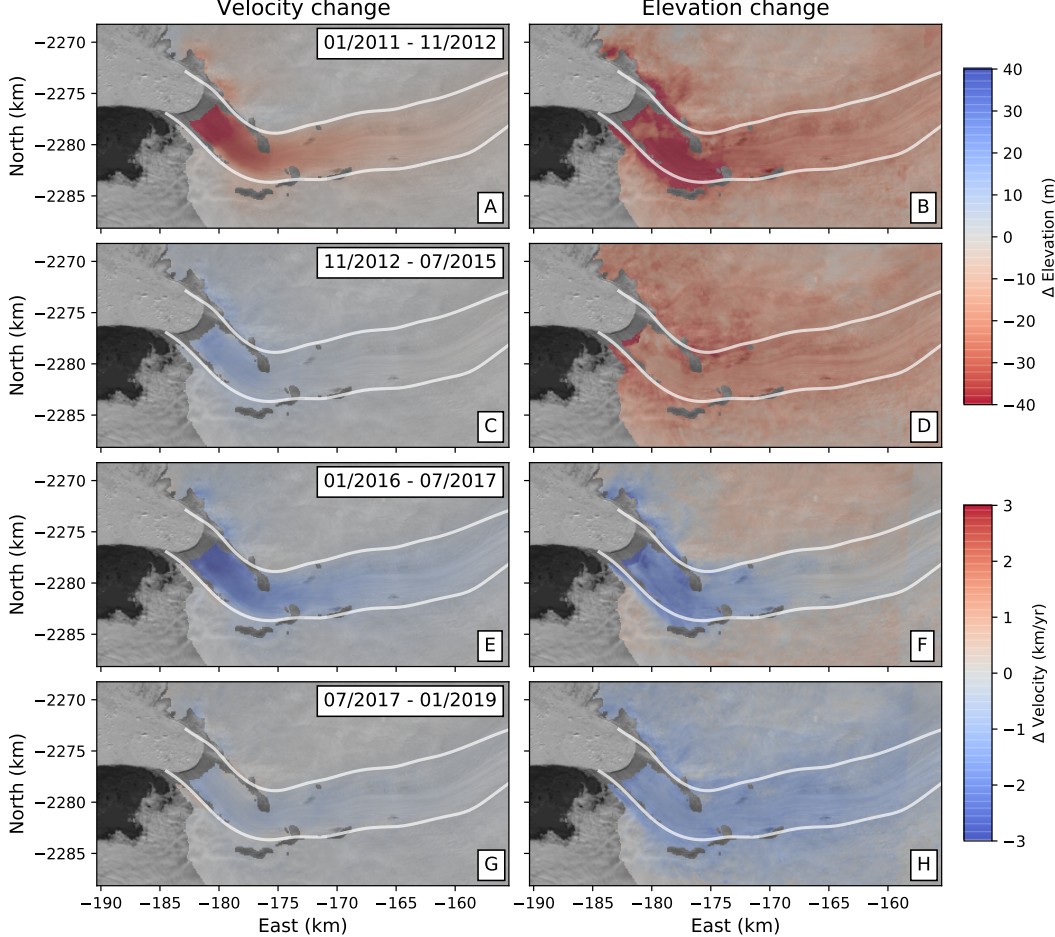

**Figure 6.** Long-term velocity and ice surface elevation changes for select time periods. (A-B) January 2011 to November 2012: The glacier is
accelerating (positive velocity anomaly) and thinning (negative elevation anomaly), with higher rates of within the glacier. (C-D) November
2012 to July 2015: Velocities show slight deceleration while the ice surface lowers over a wider area, indicating persistent surface melt. (E-F)
January 2016 to July 2017: This time period contains the strong 2017 slowdown, which is associated with ice thickening in the main trunk
of the glacier. Note the continuing surface melt signal indicated by lowering distal to the glacier. (G-H) July 2017 - January 2019: Relatively
constant glacier flow speed and more widespread increase of ice surface elevations due to positive SMB can be seen.

due to dynamic thinning on the main trunk of the glacier, while on the slower moving ice, we can observe a widespread lower-
ing of elevations. A comparison of time series for points on and off the glacier suggest that much of the ice in the surrounding

areas has been lowering since before the observation period, which is associated with a period of exceptionally high surface





melt starting around 2009. The widespread lowering of the ice surface persists even during the slight deceleration of velocities over the glacier from 2012 to 2015 (Figure 6C-D). During these years, the ice surface has likely adjusted to the initial speedup in 2012, leading to a reduction in driving stress and velocities. The 2017 slowdown is coincident with thickening of the ice on the main trunk of the glacier while the inland ice continues thinning (Joughin et al., 2018; Khazendar et al., 2019; Joughin

et al., 2020). From 2017 onwards, we observe more widespread thickening of the ice as the glacier velocities continued to decrease. Despite the recent thickening trend, elevation values are still measurably lower than they were in 2010, particularly for the ice outside of the main trunk of the glacier.

### 4.4   Calving Front Forcing

The position of the calving front has been reported to be the dominant factor influencing ice dynamics for Jakobshavn Isbræ

over observable, particularly seasonal, timescales (Nick et al., 2009; Joughin et al., 2012; Bondzio et al., 2017). The position of the calving front is heavily influenced by bed topography (Xu et al., 2013; Morlighem et al., 2016), local ocean temperatures that influence subaqueous melting of the terminus (Holland et al., 2008; Khazendar et al., 2019), and changes in melange rigidity (Joughin et al., 2008; Cassotto et al., 2015; Robel, 2017; Xie et al., 2019; Joughin et al., 2020). Changes in the calving front driven by iceberg calving can reduce back stresses upstream of the new calving front, allowing ice near the terminus to

accelerate and subsequently thin. Local thinning of the ice steepens the surface slope, thereby increasing gravitational driving stress and causing further acceleration and thinning (Joughin et al., 2012). For glaciers where the calving front is located on a retrograde bed, thinning-induced retreat of the terminus corresponds to increasing terminus ice thickness, which further increases the driving stress (Joughin et al., 2020). The redistribution of mass during dynamic thinning propagates upstream as traveling waves, with phase velocities and attenuation length scales that we constrain using the methodology presented in

this study. The thinning of the ice surface throughout the glacier is hypothesized to cause a further, indirect enhancement of velocities by causing a reduction in overburden pressure (weight of the ice column per unit area), which influences the response of the glacier to processes such as basal lubrication by drainage of surface meltwater in the summer melt season. While the role of basal lubrication has been shown to affect the seasonal cycle on certain glaciers (e.g. Howat et al., 2010; Bevan et al., 2015), the magnitude and timescale of the influence of basal lubrication on the velocities in Jakobshavn remains unresolved.

Consistent with earlier work (e.g. Joughin et al., 2012; Cassotto et al., 2019), we observe a strong correlation between the seasonal variations in ice velocity and the year to year variations of the front, which is a proxy for terminus depth as the front retreats down a reverse slope (Figure 7). The timing of maximum retreat for a given year is closely associated with the timing of peak seasonal ice velocity for a point 1.4 km upstream of the 2017 terminus position. For the four years associated with increased seasonal velocity amplitudes (2012 – 2015) during the summer, the calving front retreats past the narrow section in

the bed referenced by (Cassotto et al., 2019) (and defined as our reference position for the terminus position time series) into the wider and deeper basin (values > 0 in Figure 7C), which supports previous hypotheses that even subtle bed constrictions in the fjord can lead to large increases in ice velocity in response to terminus retreat when ice elevations are near flotation heights (Cassotto et al., 2019). In 2016, the calving front also retreats past the same pinning point, but the seasonal velocities do not reach the same peak as the previous four years. In fact, in 2016, the calving front starts the summer melt season in a



**Figure 7.** Comparison between time series of observed terminus positions and ice velocities at various timescales. A) Seasonal ice centerline velocity at a point 1.4 km upstream from the reference 2017 terminus position. Speeds are shown on an annual timescale (referenced to to April 1) and are plotted relative to the speed on April 1 of the respective year. Line colors correspond to the colormap in D, and bold lines correspond to the years 2012 – 2016 where strong amplification of the seasonal variations is observed (Fig. 1). B) Long-term ice velocity and terminus position. Velocities are extracted from the same point as in (A). Terminus positions are measured at the intersection of the time-varying calving front with the glacier centerline and are referenced to the position of the shallow pinning point highlighted by (Cassotto et al., 2019). C) Seasonal variations in terminus position with line colors and widths corresponding to lines in (A). D) Correlation between terminus position and relative velocity at the same location as in (A) and (B). Here, the points have been grouped into two temporal clusters: July 2011 – January 2017 and all other times, with the former time frame defined by the period when maximum seasonal retreat of the terminus position took it behind the reference reference pinning point. Thus, the scaling relationship between terminus position and velocities changes depending on whether the terminus has retreated beyond the pinning point. The text colors for the $R^2$ values correspond to the trend-line colors, while the point colors in (D) correspond to line colors in (A) and (B).





more retreated position, which is a consequence of the front not sufficiently advancing in 2015 and prematurely retreating in December of that year. Thus, while the seasonal amplitude in 2016 is less than the previous four years, the absolute velocities are still high (Figure 1C). For the years 2009 – 2011 and 2017 – 2018, the calving front does not retreat past the pinning point, which results in seasonal velocity variations with markedly smaller amplitudes.

Following Lemos et al. (2018), we compute the correlation between the measured calving front position (relative to the
reference pinning point) and short-term velocities for the same point 1.4 km upstream of the 2017 terminus (Figure 7D). The change in the velocity response to front variations between 2012 and 2015 can be clearly seen as a distinct cluster as compared to the other years in the observation period. For these four years, a linear regression yields a coefficient of determination ($R^2$) of 0.71 with seasonal velocities scaled by 2.2 km/year per km of front retreat. For the other years, the regression results in an $R^2$ of 0.48 with a scale factor of 1.2 km/year per km of front retreat. Thus, from 2012 to 2015, the velocity response is more
strongly correlated with front position with larger peak-to-peak variations than the other years, presumably due to the retreat of the front past the reference pinning point described by Cassotto et al. (2019). The higher level of correlation for the years 2012 – 2015 is likely driven in part by the proximity of our point of comparison (1.4 km upstream of the 2017 terminus) to the retreated front position for those years. However, comparison of front position with velocities at a moving point 1-km upstream of the front still show lower correlation for the years 2009 – 2010 (Joughin et al., 2020), which underscores the importance of
bed topography on the response of ice flow to front position.

On timescales longer than a year, calving front positions and ice speeds co-vary, but the relationship is more non-linear than on seasonal timescales (Figure 7B and D). After the disintegration of the ice tongue between 1998 and 2004, the front rapidly retreated about 4 km over the period from 2004 to 2011. During this time, the ice speed near the 2017 terminus increased about 1.5 km/yr, half of the roughly 3 km/yr increase associated with the 2012 speedup. The 2012 speedup on the other hand
coincided with a 2 km retreat of the calving front (Figure 7B).

## 5  Discussion

Decomposition of the time-dependent velocity and surface elevation fields into distinct temporal scales reveals a repeating pattern on Jakobshavn where velocity and surface elevation variations originate at the terminus in response to changes in calving front position. The coincidence between speedup and slowdown of the glacier with thinning and thickening, respectively,
suggests a dynamic origin to the physical mechanism generating these variations. In this section, we detail the observed wave phenomena, introduce the proposition that the observed traveling waves are kinematic waves, and discuss possible paths for future development of observational methods that will enable progression toward robust and efficient techniques for fusing remote sensing data from multiple sources and using *in situ* observations as prior information to constrain the inversions.

### 5.1  Wave phenomena

Our results indicate that velocity variations initiating at the terminus of Jakobshavn propagate upstream as traveling waves with frequency-dependent propagation speeds (phase velocities) and attenuation length scales. To our knowledge, ours are the first



results to explicitly quantify wave propagation at seasonal and multi-annual timescales using remote sensing observations, and, importantly, to show that traveling waves in this range of frequencies are dispersive, meaning that phase velocity is a function of frequency. These results on Jakobshavn complement our inferences of wave propagation for hourly to fortnightly timescale variations in the flow speeds of Rutford Ice Stream, Antarctica, using remote sensing data (Minchew et al., 2017), helping to demonstrate the largely untapped potential of time-dependent remote sensing observations to quantify wave phenomena. Our ability to quantitatively observe wave propagation in glaciers using remotely sensed observations adds a new class of information and unique constraints on the mechanics of glacier flow — most notably the rheology of the ice-bed interface (*i.e.*, the form of the sliding law) and the rheology of natural glacier ice — for the simple reason that these mechanics influence both the state of any given glacier as well as the response of the glacier to external forcing. At the moment, data sparsity limits our ability to quantify wave propagation to phase velocities and attenuation length scales. As more data become available, the time series methods outlined above should allow for observations of waveforms manifest in surface elevation fields and broader and more refined constraints on dispersion relations (the relationship between frequency and wavelength) for individual glaciers. Realizing this potential for remote sensing time series is important because the characteristics of wave propagation, specifically the dispersion relation as defined for a wide range of frequencies, are intrinsic properties of dynamical systems, if we define the system in this case such that it includes the glacier and boundary conditions.

### 5.1.1 Observations of wave propagation

The most important and novel observables from this study are the phase velocities and attenuation length scales for seasonal and multi-annual velocity variations in response to changing calving front position. Notably, both the phase velocities and attenuation length scales are frequency-dependent, with the higher frequency seasonal signals propagating faster but not as far as the multi-annual signals. These results indicate that the traveling waves we observe, which appear to have a common source in the displacement of the calving front, are dispersive. This observation of dispersive wave propagation enables rich diagnostic tools to understand glacier mechanics, which we will explore in future work.

Physical insight can be gleaned from the absolute values of phase velocity (Fig. 3). One key take-away from the absolute values are that the phase velocities we observe are everywhere at least an order of magnitude faster than the local mean (downstream) flow speeds of the glacier. We cannot estimate the fastest wave propagation speeds, which occur in the 5 km immediately upstream of the 2017 terminus position, because of the limited temporal resolution of the data. However, we estimate the lower bound on the phase velocity in this region based on the timing of the observations to be around 500 m/day (182.5 km/yr), meaning that the traveling waves we observe propagate upstream at least 18 times faster than the local mean glacier flow speed. In the region 5–8 km upstream there is a reduction in phase velocities, and from 8–20 km upstream we find the slowest and best constrained phase velocities at all frequencies. The propagation of these seasonal traveling waves is likely influenced by other indirect effects in the later part of the season, such as changes in overburden pressure due to ice thinning which can effect the response of ice flow to basal lubrication (Joughin et al., 2012).

The slowest phase velocities we report are for the multi-annual transient events that consist of a speedup centered in 2012 and a slowdown centered in 2017. Both the 2012 speedup and 2017 slowdown have approximately the same phase velocity,



which we find to be $231 \pm 16$ m/day ($84 \pm 6$ km/yr). Thus, whether associated with retreat (glacier speeds up) or advance (glacier slows down) of the terminus position, multi-annual signals propagate upstream roughly seven times faster than the mean flow speed near the terminus of Jakobshavn, or more than an order of magnitude faster than the local mean flow speeds. Taking both of the observed multi-annual signals to have periods of 3 years, we estimate the wavelength of the traveling waves

to be $2\pi c_p/\omega \approx 252 \pm 18$ km, where $c_p$ is the phase velocity and $\omega$ the angular frequency. At $\sim$50 times the width, $\sim$200 times the thickness of the glacier near the terminus, and $\sim$5 times the length of the glacier, the wavelength of observed multi-annual waves is much longer than any spatial dimension of the glacier. For seasonal frequencies, we observe phase velocities in the upstream 8–20 km section of $398 \pm 20$ m/day ($146 \pm 7$ km/yr), which is more than an order of magnitude faster than the glacier flow speed near the terminus. Taking the period of seasonal variations to be 1 year, we estimate the respective wavelength to

be $\approx 146 \pm 7$ km. At $\sim$30 times the width and $\sim$100 times the thickness of the glacier near the terminus, the wavelength of seasonal variations is much longer than typical characteristic length scales for glacier flow but shorter than the multi-annual-period waves by a factor of approximately $1/\sqrt{3}$, as discussed below. The attenuation length scales we report for both seasonal and multi-annual-period traveling waves are more than an order of magnitude shorter than the wavelengths, consistent with previous studies of kinematic waves that found diffusive behavior with attenuation timescales shorter than their periods (Lick,

1970; Jóhannesson, 1992; Gudmundsson, 2003).

The wavelengths of the observed waves are perhaps clearer when considered in terms of the phase velocities. Using the same periods, we see that the ratio of phase velocities is approximately equal to the square root of the ratio of the frequencies, such that

$$\frac{c_p^{multi-annual}}{c_p^{seasonal}} \approx \sqrt{\frac{\omega^{multi-annual}}{\omega^{seasonal}}} \approx \frac{1}{\sqrt{3}} \qquad (10)$$

in agreement with the estimates of a wavelengths just discussed. This estimate places useful constraints on the form of the dispersion relation, which we will explore in future work. We note that the observed attenuation length scales appear to deviate somewhat from the square-root-frequency relation as in Eq. 10, with the attenuation length scale for the seasonal signal being approximately half that for the multi-annual signal (Fig. 3). As with the phase velocities, this relationship between attenuation and frequency provides useful constraints on the dispersion relation.

Given the estimates of phase velocity at two different frequencies, it is desirable to estimate the group velocity, defined as the rate at which the envelope of a wave packet travels, and thus the rate at which the energy of the wave packet travels. Mathematically, group velocity is defined as $c_g = \partial \omega / \partial k_r$, where $k_r = \omega / c_p$ is the real component of the angular wavenumber. Our estimates of group velocity are crude given that we only observe waves with two frequencies and the period of the multi-annual signal is not well-defined because we use an integrated B-spline as a smooth step function to fit the transient.

Nonetheless, assuming, as before, that seasonal signals have a period of one year and the observed multi-annual signals both have periods of three years, we estimate a group velocity of $c_g \approx 230$ km/yr, which is faster than the respective phase velocities. Therefore, in this relation between phase and group velocities, the waves we observe are analogous to capillary waves, though the analogy goes no further as the physical mechanisms governing capillary waves are not all applicable to the low-Reynolds number regime of glacier flow.





The wave speeds we observe on Jakobshavn are appreciably faster than those observed on Mer de Glace, France, even when
we correct for the differences in mean flow speed between the two glaciers. Traveling wave speeds on Mer de Glace following
numerous perturbations in surface mass balance were synthesized by Lliboutry and Reynaud (1981) to be in the range 450–725
m/yr, or about 4–6 times the mean flow speed of the glacier. The traveling waves we observe on Jakobshavn are moving two
orders of magnitude faster upstream than traveling waves on Mer de Glace. The ratio of phase velocity to local mean flow
speed is roughly a factor of two higher on Jakobshavn than Mer de Glace. There are marked differences in these two glaciers
and the perturbing forces are quite different — retreat of the terminus on Jakobshavn and perturbation in surface mass balance
on Mer de Glace — but the qualitative differences in observed wave propagation speeds are noteworthy.

When comparing the results presented here with observed wave propagation on other glaciers, we note dramatic differences
between the phase velocities and attenuation length scales on Jakobshavn and wave propagation related to much higher fre-
quency (14.76-day period) variations in ice-surface velocity that we previously reported for Rutford Ice Stream, West Antarc-
tica (Minchew et al., 2017). On Rutford — with a mean flow speed near the grounding line of approximately 375 m/year
— downstream variations in buttressing stresses over the ice shelf, grounding line position, and/or water pressure at the bed
(Gudmundsson, 2006, 2007; Rosier et al., 2014, 2015; Minchew et al., 2017; Robel et al., 2017; Rosier and Gudmundsson,
2020) drive variations in the flow speed of Rutford that propagate with a phase velocity of approximately 24 km/day for the
first 40 km upstream of the grounding line, and then at a faster rate of 34.3 km/day further upstream. Thus, observed waves on
Rutford driven by ocean tides propagate two orders of magnitude faster than those we observe on Jakobshavn. The attenuation
length scales are also markedly different: approximately 45 km on Rutford versus 7 km (seasonal) and 14 km (multi-annual)
on Jakobshavn. The marked differences in forcing frequencies and propagation speeds and distances suggest fundamental
differences in wave types and mechanisms.

### 5.1.2   Kinematic wave propagation


We hypothesize that the traveling waves we observe can be classified as kinematic waves because of their long periods (months
to years), strong correlation with calving front position, and marked dynamical thinning of the glacier corresponding to in-
creases in velocity. Kinematic waves represent a special case in a broader spectrum of wave behavior that includes various
forms of dynamical waves. As the name suggests, dynamical waves are driven primarily by pressure and stress gradients and
arise in a variety of contexts, such as seismic waves, flexural waves (Lipovsky, 2018), shallow-water waves, so-called "sea-
sonal waves" on glaciers (Fowler, 1982; Hewitt and Fowler, 2008), and the response of outlet glaciers to ocean tides (Rosier
et al., 2015; Rosier and Gudmundsson, 2016; Minchew et al., 2017). On the other hand, kinematic waves arise primarily from
the redistribution of mass, with propagation characteristics dominated by mass conservation rather than momentum balance.
Kinematic waves have been used to model phenomena as varied as glacier surges, river floods, and traffic flow (Lighthill and
Whitham, 1955; Nye, 1958, 1960). All waves on glaciers will be driven by some combination of dynamic (momentum bal-
ance) and kinematic (mass balance) processes, and the dominance of one driving mechanism over another informs the broad
characteristics of the full dispersion relation for any given glacier at any given time.





Our hypothesis that the waves we observe are kinematic waves is consistent with previous studies that explore various aspects of kinematic wave propagation and the response of tidewater glaciers to changes in terminus position, from observations (Howat

et al., 2005; Joughin et al., 2008; Felikson et al., 2017) to flow models (Thomas, 2004; Pfeffer, 2007; Nick et al., 2009; Joughin et al., 2012; Williams et al., 2012). Often, these arguments are based on the logical conclusion of progressive draw-down of the glacier surface. In essence, this process plays out such that changes in the terminus position through calving locally alter longitudinal (normal) stresses, allowing the glacier to locally accelerate. The local acceleration alters longitudinal stresses and increases flux divergence, causing the glacier to thin. Localized thinning steepens the surface slope, thereby increasing

gravitational driving stress and generating upstream acceleration (Joughin et al., 2012). The process repeats as the surface is progressively drawn down and a wave propagates upstream. Similar mechanisms operate in reverse for re-advance of the terminus position. This explanation is broadly consistent with our observations showing that changes in surface elevation are coincident with changes in surface velocity (Fig. 6). We reserve for future work a detailed exploration of wave propagation in glaciers, including tests of the kinematic wave explanation of the observations we present in this study.

However, we note two important caveats to the applicability of the kinematic wave description to our observations. The first is that the wave speeds we observe are an order of magnitude faster than the local mean flow speeds, making them much faster than kinematic wave speeds proposed previously (Nye, 1960; Lliboutry and Reynaud, 1981; Weertman and Birchfield, 1983; van de Wal and Oerlemans, 1995). This discrepancy likely arises from a combination of the assumptions made in previous estimates and the fact that our observed phase velocities are not corrected for diffusivity, and so are not direct measurements

of the kinematic wave speed (Lliboutry and Reynaud, 1981). As an example of the assumptions made in earlier estimates of kinematic wave speeds, Nye (1960) assumed a shallow-ice approximation for the flow regime (wherein gravitational driving stress is balanced locally at the bed) and a power-law relation between slip-rate and drag at the ice-bed interface (*i.e.,* the sliding law) with values for the exponent defined by Weertman (1957) for regelation and viscous flow of ice around roughness features in a monochromatic bed. Such a sliding law does not account for the full range of possible sliding mechanisms (Schoof, 2005;

Joughin et al., 2019; Zoet and Iverson, 2020; Minchew and Joughin, 2020), meaning that estimates from Nye (1960), and others that follow the same basic formulation, do not capture the range of possible values for kinematic wave speeds. More work is needed to reconcile our observations with models of traveling wave propagation.

The second caveat to our conclusion that the waves we observe are kinematic waves is that our observations indicate that the observed waves are dispersive. Recall that we define dispersion in the traditional sense to mean "frequency dispersion" — the

frequency dependence of wave speed — whereas some authors, notably Lighthill and Whitham (1955), use the term "amplitude dispersion" to describe the amplitude dependence of wave speed. The latter is a general characteristic of kinematic waves that can cause waveforms to change shape as faster waves overtake slower waves (Fowler, 1982, 2011). However, the amplitude dependence of wave speed does not account for our observations for three reasons. First, the phase velocity of waves with seasonal periods are relatively consistent despite large differences in the amplitude of seasonal variations (Fig. 2). Second, the

multi-annual variations in glacier flow speeds have comparable amplitudes to the seasonal variations and yet propagate slower by a factor of $\sqrt{3}$, nearly equal to the square root of the ratio of frequencies (Eq. 10 and Fig. 3). Third, the observed phase velocities for the 2012 speedup and 2017 slowdown propagate at nearly the same speed (Fig. 3) despite different amplitudes of





variations in surface flow speed (Figs. 1 and 2) and elevation (Fig. 6). Thus, we conclude that differences in amplitude cannot explain the disparities in phase velocities from waves with seasonal and multi-annual periods. Our observations, therefore,
provide evidence of (frequency) dispersion in traveling waves.

Evidence for dispersion is noteworthy in part because kinematic waves on glaciers are governed by the continuity equation, which can be expressed in the form of the first-order wave equation; kinematic waves are, therefore, non-dispersive in the classical theory (Lighthill and Whitham, 1955; Nye, 1960). More recent work by, for example, Pfeffer (2007) and Felikson et al. (2017) present models for tidewater glaciers that are similar to the classical kinematic wave models and are likewise non-
dispersive. While we leave for future work a detailed theoretical model accounting for dispersion in waves with seasonal to multi-annual periods, we speculate on a few possible sources of dispersion. The first candidate for the source of dispersion can be broadly defined as effects from longitudinal (extensional) normal stresses on wave propagation. If important, longitudinal stresses would entail inclusion of dynamical processes operating with greater or equal importance to the kinematic wave processes. As mentioned previously, kinematic waves are special cases in a broader spectrum of wave phenomena that must
include various types of dynamical waves, and we should expect there to be a range of frequencies in which both mass and momentum balances play non-negligible roles in wave propagation.

The rheology of ice is another candidate for dispersion. We may intuitively expect waves with periods comparable to the viscoelastic relaxation time (ratio of dynamic viscosity to shear modulus) to be dispersive as the combination of viscous and elastic responses mean that waves of different frequencies will travel through ice with with different effective rheologies. In
this case, the restoring force provided by elasticity is proportional to strain, which will vary with wavelength. However, the periods of waves we observe are much longer than the viscoelastic relaxation time — which is typically of order hours to weeks (Gudmundsson, 2007) — meaning that ice is essentially a purely viscous fluid over the timescales relevant for our observations, and therefore elasticity is an unlikely source of dispersion. The non-Newtonian viscosity of ice is also unlikely to generate dispersion in kinematic waves due to the long periods of the wave relative to the viscoelastic relaxation time, which
is supported by previous studies of kinematic waves (Nye, 1960; van der Veen, 2001; Pfeffer, 2007; Felikson et al., 2017).

Other obvious physical processes that could give rise to dispersion in traveling waves with seasonal and multi-annual periods are related to water pressure at the bed. In the interest of brevity, we divide the relevant processes into two broad categories: subglacial hydrology and till porosity. For subglacial hydrology, the well-known dependence of the hydrological pressure gradient on the surface slope of the glacier may be an important consideration (Flowers, 2015). After all, kinematic waves
propagate upstream due to a progressive draw down of the surface, which entails a transient steepening of the surface slope. The response of the subglacial hydrological system to the transient changes in glacier geometry are beyond the scope of this study and are worthy of further consideration in future work. As another example, dilation and compaction of subglacial till has recently received attention as a possible mechanism for triggering surges in glaciers with deformable beds (Minchew and Meyer, 2020) and has long been considered a potentially important mechanism in the centennial timescale dynamics of ice
streams (Tulaczyk et al., 2000; Robel et al., 2016; Meyer et al., 2019b). In this mechanism, till dilation is influenced by the overburden pressure at the bed (*i.e.,* the weight of ice per unit area) and the change in slip rate at the bed. As the surface velocity increases, it is likely that a corresponding increase in slip rate manifests at the bed and can be expected to cause the





deforming till layer to dilate. For realistic values of hydraulic permeability of the till, dilation will cause a temporary decrease in pore water pressure. Minchew and Meyer (2020) showed that this processes can (in an idealized model) lead to glacier surges by delaying the evolution of the till to a new steady-state. In a related sense, dilation and the subsequent evolution of pore water pressure provide a possible mechanism for altering the mechanical properties of the glacier bed in such a way that might generate dispersion in waves with seasonal and multi-annual periods.

## 5.2   Future work in remote sensing

The increasing availability of surface velocity and elevation fields sampled at monthly-to-sub-monthly timescales will continue to provide opportunities to study the rapid evolution of fast-flowing glaciers to various environmental forcings. The operational capabilities of several working groups that produce velocity fields over the Greenland and Antarctic Ice Sheets will consistently improve as new data are made available and techniques for generating velocity estimates are refined. In particular, the upcoming NASA-ISRO Synthetic Aperture Radar (NISAR) mission will generate unprecedented volumes of data that are useful for quantifying surface change for a number of scientific applications, including glacier dynamics (NISAR, 2018). The wide imaging swath (~240 km) coupled with a 12-day repeat cycle and global coverage will allow for systematic observations of high-resolution velocity variations over interconnected glacier networks and coupled ice stream and ice shelf systems. Such observations will facilitate quantification of the spatiotemporal responses of glaciers and ice streams to any changes to the stress state, such as changes to the terminus position, loss of ice-shelf buttressing, changes in frictional properties of the bed, evolution of the subglacial hydrology. These processes will likely result in wave phenomena similar to those observed at Jabovshavn Isbræ (this study) and Rutford Ice Stream (Minchew et al., 2017) and would be well-observed with platforms like NISAR. Furthermore, the quantification of phase velocities and attenuation length scales at multiple forcing frequencies would provide valuable constraints on a general theory for wave propagation for fast-flowing glaciers because the characteristics of wave propagation are intrinsic properties of any given glacier system, which includes the boundary conditions.

The temporal resolution of surface DEMs is currently a limiting factor in quantifying sub-annual dynamical thinning. In this study, we noted that the thinning signal in the ice adjacent to the fast-flowing regions may be due to oversmoothing of the time series due to the limited temporal resolution of the ArcticDEM dataset. Therefore, elevation or altimetry datasets that have increased temporal sampling, such as IceSat-2, may help isolate the shorter-term dynamic signals from any longer-term SMB-based variations. In particular, future analysis would benefit from the 91-day repeat time of IceSat-2 for capturing seasonal elevation variations for direct comparison and synthesis with the velocity seasonal variations. This type of analysis could lead to a full three-dimensional velocity time series, which has the potential to improve quantification of strain and stress fields, constraints on ice rheology, and assimilation of velocity data into state-of-the-art ice flow models. For glaciers and ice streams where a persistent ice shelf or tongue exists, tidal forcing may become an important stress perturbation, in which case accurate reconstruction of vertical displacements would be necessary in order to constrain the dominant tidal constituents of motion (Minchew et al., 2017).

The flexible time series framework described here introduces the potential for using *in situ* observations as prior information (encoded in the prior model covariance matrix $C_m$) in forming time-dependent surface velocity fields. One example of this





synergy between remote sensing and *in situ* data is the use of GPS/GNSS observations to constrain the form of the temporal basis functions, as we did in Minchew et al. (2017). Another, less obvious, example is the potential for employing catalogs of calving events gleaned from seismic observations (Olsen and Nettles, 2017, 2019; Olinger et al., 2019) to constrain the

timing and duration of transient accelerations in ice flow. Such constraints on the temporal evolution of the fields observed from remote sensing observations should afford novel opportunities to constrain phenomena such as the localization of strain rates (and, thereby, stresses) associated with fracture and calving. We expect the usefulness of the flexible methods we present here to grow as more remote sensing and *in situ* data become available.

## 6   Conclusions

We have presented a framework for forming continuous time-dependent surface velocity and elevation fields from publicly available surface velocity and elevation data. This framework is based on a sparsity-regularized linear regression method that reconstructs time series as a linear combination of relevant basis functions. The flexibility and expressive power of the basis function representation allows for accurate reconstruction of time series in the presence of noisy and missing data while also allowing for a natural decomposition of the total signal into signals of multiple temporal scales. Over Jakobshavn Isbræ, this

decomposition permitted a detailed investigation into the spatiotemporal characteristics of the evolving seasonal cycle of ice speedup and slowdown in response to terminus variations. Analogously, longer-term changes in velocity were isolated and shown to also be primarily driven by longer-term terminus variations. This type of analysis is directly applicable to many outlet glaciers in Greenland, Antarctica, and other areas where multitemporal remote sensing data is available and could improve our understanding of the dynamic response of glaciers to various geographic and environmental forcings.

We demonstrated that the time series reconstruction permitted the quantification of traveling wave propagation resulting from terminus forcing functions at different temporal frequencies. These results build upon an important new area of research that aims to achieve a mechanistic understanding of glacier flow from time-dependent velocity data. To our knowledge, our results are first to show from observations that waves on glaciers with seasonal to multi-annual periods are dispersive, with a ratio of observed phase velocities approximately equal to the square root of the ratio of frequencies. We hypothesize that the

observed waves can be classified as kinematic waves based on their long periods (much longer than the viscoelastic relaxation time), correlation with changes in the terminus position, and coincident variations in surface velocity and elevation. These observations of traveling waves are only possible due to the strong velocity response to changes in terminus position, as well as our ability to isolate short- and long-term signals in the velocity data. Looking forward, we aim to assimilate other velocity sources for Jakobshavn Isbræ (e.g., optical or Sentinel-1), as well as other elevation and altimetry data sets to improve temporal

sampling and to obtain full 3D surface velocity time series. The resultant dataset will likely lead to a marked improvement in incorporating velocity data in ice flow models for simulation and inversion of mechanical properties.

*Code availability.*   The time series analysis and decomposition tools are available at https://github.com/bryanvriel/iceutils.



*Data availability.* The velocity data are available at NSIDC (NSIDC-0481 at https://nsidc.org/data/measures/gimp, last access: Sep 2019). The original elevation data are available from the Polar Geospatial Center, University of Minnesota (ArcticDEM at https://www.pgc.umn.edu/data/arcticdem, last access: Sep 2019). The pre-2017 calving fronts obtained in this study are available from https://doi.org/10.1594/PANGAEA.897066 (Zhang et al., 2019). The post-2017 calving fronts delineated from TerraSAR-X data are available at the University of Washington Research Works Archive at https://doi.org/10.6069/XQS7-CD47 (Joughin et al., 2020). Landsat-8 imagery are available from EarthExplorer (https://earthexplorer.usgs.gov, last access: June 2020). The bed elevation (BedMachine v3) is available at NSIDC (https://nsidc.org/data/IDBMG4, last access: Sep 2019).

*Author contributions.* B.R. and B.M. conceived the project, analyzed the data, and wrote the manuscript. I.J. processed the GIMP data and provided valuable insights and feedback on the manuscript.

*Competing interests.* The authors declare no competing interests.

*Acknowledgements.* The authors thank Jerome Neufeld, Ian Hewitt, Denis Felikson, and Colin Meyer for insightful discussions. B.R. and B.M. were partially funded through NSF award OPP-1739031.



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
