# Peer review of "Observing traveling waves in glaciers with remote sensing: New flexible time series methods and application to Sermeq Kujalleq (Jakobshavn Isbræ), Greenland"

_The Cryosphere, 2020_

## Referee Comment (RC1) · Anonymous Referee #1 · 4 Sep 2020

— General comments —

Using high-temporal-resolution remote sensing of glacier velocity, this paper presents observations of wave propagation along Jakobshavn Isbrae, Greenland. The sampling of the data and the sparsity-regularized linear regression method that the authors employ allow the decomposition of velocity variation signals into two frequency categories: seasonal and multi-annual. This allows the authors to, for the first time measure wave propagation speeds for different forcing frequencies, showing wave dispersion along glacier flow. The paper is very thorough and complete, present-

ing information in multiple ways, which helps the reader to fully grasp the interpretation. Because the results in this paper are very novel and cutting-edge, I appreciate the thoroughness with which the authors provide interpretation, additional hypotheses to be tested in future work, and ideas for how future remote sensing can shed light on the concepts presented. I really enjoyed reading the paper for this aspect in particular, as it helped frame the results that are presented and sparked ideas for potential future work and questions to be answered. The authors should also be commended for putting together an excellent tutorial with an example (https://github.com/bryanvriel/iceutils/blob/master/doc/time_series_inversion.ipynb). I found this resource very useful as I was reading the methods described in the paper.

In my assessment, I found no major flaws in the manuscript. I present five general comments and suggestions here and more detailed line edits below.

1.) I would like more description of how the B-splines are constructed in the methods section (lines 101-116). I am not an expert on this approach to time-series analysis and the description provided may be enough for someone with a deeper background. But, for the non-experts, I suggest adding a couple of sentences to explain, in plain language, that there are a set of seasonal B-splines (with period of 1 year) and a set of transient B-splines (with period of < 1 year) that are simultaneously being fit to the observed velocities, if that is indeed the case. This would then connect nicely with the paragraph describing how the data is detrended on lines 129-137. This is my interpretation and, without looking at the tutorial code, I'm not sure that I completely understand how the B-splines are constructed. For example, are the seasonal B-splines fit to data that falls within a window of 1 year? This is the kind of thing that would clarify for me, the non-expert, how this approach works.

2.) At times, I found it a bit confusing tracking what velocity quantity was being discussed (i.e., seasonal velocity, long-term velocity, etc.). I would like to suggest two ways to address this. First, I suggest adding to the methods section (probably to line 137) a statement such as, "Throughout the paper, reference to seasonal velocity represents the quantity d_s and references to long-term velocity represents the quantity ..." Second, I ask the authors to carefully go through the entire manuscript to ensure that all references to "seasonal velocity" and "long-term velocity" do, in fact, refer to these exact quantities. I appreciate that the authors have likely already done this and I commend them for their writing, which is already for the most part very clear. I only ask that a final pass is done through the text before the resubmission to double-check the references to these different quantities of velocity.

3.) Along the lines of future work, I would like the authors to add some brief discussion on the use of in-situ measurements to measure the propagation of waves. For example, can terrestrial radar or laser scanners be used to provide high-temporal-resolution measurements that can help further constrain glacier waves? This can be added to the end of Section 5.2.

4.) In a few parts of the discussion, it is stated that velocity and surface elevation are responding to changes in calving front position but this causality is not shown by the results of the paper. It is shown definitively that variations initiate at the terminus and propagate upstream and that these variations are well-correlated with terminus motion. However, causality (in one direction or the other) is not shown by the analysis here. The language surrounding this discussion should be revisited and revised. Perhaps I have missed something and this causality can be inferred but, in that case, it needs to be made more explicit and clear in the discussion. Otherwise, the causality wording should be changed to discussing the correlation between terminus position and velocity.

5.) Finally, a minor comment that applies throughout the paper. There are a couple of places where the "southern bend" of the glacier is mentioned and I suggest adding something that indicates this region to all of the map-view figures.

— Minor comments —

[line 50] What are sub-epoch velocity changes?

[line 137] This sentence can be removed.

[Fig 1] State in the caption that the map coordinates are polar stereographic north (EPSG:3413).

[Fig 1] I suggest replacing the manually drawn white lines in panel A with either the calculated contours of maximum shear strain rate or with the contour where the bed is at sea level. This would be a more accurate depiction of the trough and the main trunk of the glacier.

[Fig 1] The sentence "Mean velocities are added to time series for visual clarity" does not make sense to me. How are the mean velocities depicted in the plots? And what are these means (spatial? temporal?)?

[Fig 1] Clarify how the data is detrended in C and D. For example, something like: "white dots indicate (B) observed speeds, (C) observed speeds detrended using seasonal splines, and (D) observed speeds detrended using seasonal and transient splines."

[Fig 2] The approximate solid black lines drawn are very helpful in illustrating wave propagation and it is clear from the differences in panels A and B that phase speeds of seasonal signals are much different from multi-year signals. However, I would like to see calculated contours drawn on each panel. These could be the zero contours or any other arbitrary value and they can be displayed in grey, with the approximate lines in darker black for illustrative purposes.

[line 269] I am confused by the phrase "long-term signals removed." Is removing the long-term signal the same as combining the transient and seasonal signals? In other words: $d\_L = d - d\_T - d\_S$, where $d\_L$ is the long-term signal shown in Fig. 1C $d - d\_L = d\_T + d\_S$ If this is the case, I suggest rewording this from "the velocity data from 2011 to 2018 at each pixel with the estimated long-term signals removed, $d\_S$" to "the combined seasonal and transient modeled signal, $d\_T + d\_S$"

[line 300] I would replace "classical" with "time-series".

[Figure]

[line 314] This is the only place in the paper where Fig 3C is referenced and I think it is completely OK to hypothesize about the connection between phase velocity and thickness/bed but, because a figure is provided, I would suggest expanding on this a bit. Please add a sentence that explicitly states the hypothesis about the relationship between these two variables (e.g., higher/lower velocity in thicker/thinner ice).

[lines 332-334] This sentence is accurate but the "while" clause does not make sense to me. I am reading it as "along the trunk there is lowering, while on the slower ice, there is lowering." Please clarify. Perhaps this sentence is meant to say that there is a confined region of high thinning along the trunk and near the front, while on the slower ice there is still thinning but lower magnitude.

[lines 334-336] This sentence makes two claims without providing evidence. First, that the slower ice was thinning before the observation period. Second, that high melt started in 2009. Both of these must be backed up with either a figure or a citation.

[line 360] I would find it helpful to distinguish the results presented in this paper from earlier work here. Adding a clause to this sentence such as, "Consistent with earlier work ..., but at a higher-temporal resolution, we observe ..." or "Consistent with earlier work ..., but using our novel method that is better able to isolate seasonal signals, we observe ..."

[lines 360-361] This paragraph and the corresponding figure describes the relationships between (1) seasonal terminus positions and seasonal velocity variations and (2) long-term terminus positions and long-term velocity variations. Thus, I suggest rewording this sentence from "we observe a strong correlation between the seasonal variations in ice velocity and the year to year variations of the front" to "we observe strong correlations between variations in ice velocity and variations of the front at both the seasonal and long-term time scales"

[Fig 7] I suggest using a sequential colorscale, rather than a divergent one, to represent different years. The current colorscale makes it impossible to distinguish 2009 from

2018.

[Fig 7] In panel D, in addition to coloring the points according to year, distinguish the two groups separated by terminus position using different symbols (e.g., circles and squares).

[Fig 7] In panel D, it is not clear to me how the seasonal velocity variation quantity is calculated. Please add this to the text or the caption.

[lines 392-395] Strictly speaking, the results do not show that velocity and surface elevation variations are changing in response to changes in the calving front position. They are certainly correlated but causality one way or another has not been shown here. I suggest rewording this.

[line 410] I would add the word "transient": "... as well as the transient response ..."

[lines 410-411] This sentence is a bit confusing to me. What is meant by quantifying "wave propagation to phase velocities and attenuation length scales? Does this mean quantifying the relationship between wave propagation distance(?) to phase velocities and attenuation length scales? Something seems to be missing here.

[lines 412-413] Please reword to be more explicit about what is meant by "broader and more refined constraints". Does broader mean for more glaciers or at more frequencies? Does more refined mean smaller uncertainties? If so, state this explicitly.

[line 489] Can anything more be added about the kinds of waves that are observed on Rutford? Did previous work categorize what kind of waves those are? If so, I would add that here to enhance the contrast between the kinematic waves on Jakobshavn and the other type of wave on Rutford.

[line 597] "IceSat-2" should be changed to "ICESat-2"

[line 631] Please add a sentence describing, briefly, the caveats to the conclusion that the observed waves are kinematic in nature. These caveats are very nicely discussed

in detail in Section 5.1 and I think they need to be summarized in the conclusion.

[Data availability] Please add the DOI for the OMG DEMs (https://doi.org/10.5067/OMGEV-GLNA1)

---

## Referee Comment (RC2) · Anonymous Referee #2 · 17 Oct 2020

Review of "Observing traveling waves in glaciers with remote sensing: New flexible time series methods and application to Sermeq Kujalleq (Jakobshavn Isbræ), Greenland" by Riel et al.

Riel et al. present a thorough framework for creating temporally uniform velocity time series from sparse/noisy observations that effectively preserve underlying signals over multiple timescales (seasonal to multiyear). From this method of time series decomposition, velocity variability is compared to other dynamic variables (i.e. changes at the front), allowing for robust interpretations of glacier properties and driving geophysical

mechanisms. The study uses B-splines to separate multiyear signals from seasonal variability and shows how various properties of seasonal variability and signal propagation vary between distinct speed up and slow down events at Jakobshavn Isbræ. The authors then combine velocity observations with concurrent changes in elevation and suggest that the observed traveling waves are kinematic in nature, given how dynamic thinning accompanied increases in multiyear velocity and seasonal amplitude. The manuscript is very well-written and is presented in a logical manner, with the description of the framework/tools first presented, followed by a practical implementation of the technique and finally physical interpretations/limitations. The study is well motivated, and the robust framework described has potential to improve how we treat and interpret time series to better resolve the various underlying dynamic signals and understand how glaciers respond to various perturbations over multiple timescales. I found no major scientific/methodological flaws in the manuscript and think it would be a valuable contribution to TC. However, I did find some elements confusing and would like to see several concepts more thoroughly described/addressed. I list the more pertinent points first below, followed by minor and technical points.

Main Comments

My main criticism is that it was difficult to follow why and how different time periods were used for various analysis throughout the paper. I would like to see either more coherence in selected time periods used, or more description up front in the introduction/motivation to explain why various subsets of the time period are used at different points throughout the text. For example, the abstract and introduction refer to a 2009-2019 decadal study, shown in completeness Figure 1. However, Figure 2 then only shows 2011-2018, and subsequent analysis average similarly segmented time periods. As another example, mean time of peak seasonal velocity and phase velocities seem to be computed using only 2011-2018 velocity data.

Reference point used for correlation analysis: Why is a point 1.4 km upstream of the pinning point used specifically for comparison to velocity? I see that you found the highest temporal coherence between maximum retreat and maximum seasonal velocity at this location, but it would be helpful to have more information on how this coherence was derived, and why then it serves as an ideal reference point.

Please also include a map view of the reference point and pinning point on the map. I found it hard to follow why sometimes the point 1.4 km upstream of 2017 front was used, versus the pinning point, as reference in the figures. What information is lost if, as a suggestion, the pinning point is used as the single point of reference throughout the text?

Multiyear variations in surface elevation: It would be helpful to have more text describing the motivation for selecting the particular time epochs shown in Figure 6. Each of the 4 panels represent elevation changes over time intervals of varying lengths, from ~1.5 years to ~2.5 years. The selected intervals also exclude July 2015-December 2015. There may be a reason for this, but without more context it seems too arbitrary.

Discussion: I would like to see the discussion expanded to include considerations/limitations of this framework when applied to other glacier sites outside of Jakobshavn Isbræ (for example, in areas with notably lower SNR than Jakobshavn). Do you anticipate reduced SNR would limit the feasibility of this technique (and how may opting to enforce spatial coherency impact interpretations of phase velocity, propagation delay, etc).

Minor comments

Figure 2: Not a critique, but comment: This is a great figure that illustrates a lot of information in a concise way, clearly showing interannual variations in amplitude and inland propagation of signals from the front. The figure caption also included an excellent description of how phase velocity was extracted from the tangent angle.

Figure 3a: I suggest scaling the y-axes such that a range of the same magnitude is shown for both. This would allow the reader to quickly compare relative changes in

slope with distance between mean velocity without amplitude.

Figure 4: why are data from 2016 excluded from either group?

Figure 5: Please add a note to caption to remind reader that red lines delineate winter 2017 reference calving position.

Figure 7a and c: It is very difficult to differentiate between years 2009/2019 and 2017/2018. If keeping the same color scale is preferred, I suggest making 2009/2010 dashed rather than solid lines to make years more distinct.

Figure 7d and correlation analysis: Are the velocity values shown here (and used for correlation analysis) taken from the continuous fitted time series? If so, what is the sampling frequency from these curves (every week, every month?) Are the extracted velocity values uniformly spaced in time?

Line 334: "A comparison of time series for points on and off the glacier suggest that much of the ice in the surrounding areas has been lowering since before the observation period, which is associated with a period of exceptionally high surface melt starting around 2009." Can you include a citation for this?

Line 387: "After the disintegration of the ice tongue between 1998 and 2004, the front rapidly retreated about 4 km over the period from 2004 to 2011." Please include a citation for this, as front analysis used in this study did not start until 2009.

---

## Author Comment (AC2) · 29 Oct 2020

Our combined responses to referee comments are posted here:

https://tc.copernicus.org/preprints/tc-2020-193/tc-2020-193-AC1-supplement.pdf

---

## Author Comment (AC3) · 29 Oct 2020

Our combined responses to referee comments are posted here:

https://tc.copernicus.org/preprints/tc-2020-193/tc-2020-193-AC1-supplement.pdf

---

## Author Response (AR1)

**Response to reviewers for "Observing traveling waves in glaciers with remote sensing: New flexible time series methods and application to Sermeq Kujalleq (Jakobshavn Isbræ), Greenland" by Riel et al.**

**1    Summary**

We thank the reviewers for their constructive feedback and very useful suggestions. We provide a point-by-point response to their comments below. Our responses are in blue, and any additional or modified text in the manuscript is provided in quotations. Note that while the line numbers in the reviewer comments correspond to the original manuscript, the line numbers in our responses correspond to the modified manuscript. Also note that we have added two supplemental figures to address some specific comments by both reviewers.

**2    Response to Reviewer 1**

**Comment 1.** I would like more description of how the B-splines are constructed in the methods section (lines 101-116). I am not an expert on this approach to time-series analysis and the description provided may be enough for someone with a deeper background. But, for the non-experts, I suggest adding a couple of sentences to explain, in plain language, that there are a set of seasonal B-splines (with period of 1 year) and a set of transient B-splines (with period of < 1 year) that are simultaneously being fit to the observed velocities, if that is indeed the case. This would then connect nicely with the paragraph describing how the data is detrended on lines 129-137. This is my interpretation and, without looking at the tutorial code, I am not sure that I completely understand how the B-splines are constructed. For example, are the seasonal B-splines fit to data that falls within a window of 1 year? This is the kind of thing that would clarify for me, the non-expert, how this approach works.

We modified the paragraph to read:

"In this study, we use a combination of third-order B-splines and time-integrated B-splines ($B^i$-splines) to populate the columns of $\mathbf{G}$ (Hetland et al., 2012; Riel et al., 2014). Third-order B-splines are suitable for modeling seasonal signals with potential year-to-year variations in amplitude, as is observed in the ice surface velocity and elevation at Jakobshavn Isbræ (Joughin et al., 2010, 2018). To that end, we construct B-splines with effective durations (full-width at half maximum) of three months, spaced 0.2 years apart such that the center times of the B-splines repeat each year. This choice of timescale and spacing allows for reconstruction of complex, sub-annual behavior in the time series data. On the other hand, time-integrated B-splines, which exhibit slow-step behavior at particular timescales (similar to the sigmoid function), are useful for modeling *transient* variations. In this work, we define transient signals as any signal that is non-steady and non-periodic, which encompasses both rapid transients (e.g., speedup following a calving event)

and longer-term transients (e.g., multi-year increases in velocity due to long-term changes in air temperatures). This spectrum of behavior can be comprehensively reconstructed through a combination of $B^i$-splines of different timescales and onset times. For the Jakobshavn Isbræ data analyzed here, we target only longer-term transient signals by including $B^i$-splines with durations > 1 year in $\mathbf{G}$. Notationally, the partitioning of the design matrix can be represented as $\mathbf{G} = [\mathbf{G}_S, \ \mathbf{G}_T]$ where $\mathbf{G}_S \in \mathbb{R}^{M \times N_S}$ is the submatrix containing $N_S$ B-splines for modeling seasonal signals and $\mathbf{G}_T \in \mathbb{R}^{M \times N_T}$ contains $N_T$ $B^i$-splines for modeling transient signals. The regularized least squares approach in Equation 2 thus simultaneously estimates the coefficients for each submatrix such that $\hat{\mathbf{m}} = [\hat{\mathbf{m}}_S; \ \hat{\mathbf{m}}_T]$ where $\hat{\mathbf{m}}_S \in \mathbb{R}^{N_S \times 1}$ and $\hat{\mathbf{m}}_T \in \mathbb{R}^{N_T \times 1}$. Simultaneous estimation of seasonal and transient signals allows for underlying tradeoffs between the two signal classes to be maximally resolved by the full timespan of the time series."

**Comment 2.** At times, I found it a bit confusing tracking what velocity quantity was being discussed (i.e., seasonal velocity, long-term velocity, etc.). I would like to suggest two ways to address this. First, I suggest adding to the methods section (probably to line 137) a statement such as, "Throughout the paper, reference to seasonal velocity represents the quantity $d_s$ and references to long-term velocity represents the quantity ..." Second, I ask the authors to carefully go through the entire manuscript to ensure that all references to "seasonal velocity" and "long-term velocity" do, in fact, refer to these exact quantities. I appreciate that the authors have likely already done this and I commend them for their writing, which is already for the most part very clear. I only ask that a final pass is done through the text before the resubmission to double-check the references to these different quantities of velocity.

We have added the suggested sentence (at the suggested location): "Throughout this paper, references to short-term, seasonal velocity variations refer to $\hat{\mathbf{d}}_S$ while references to longer-term, multi-annual velocity variations refer to $\hat{\mathbf{d}}_T$."

Please note that in the paragraph shown in the response to the previous comment, we added a definition of transient signals to explicitly state that we are targeting longer-term, multi-annual transients in this work. Additionally, after going through the manuscript, we have made the following minor modifications to remind the reader which velocity quantity is under discussion:

Line 243: added $\hat{\mathbf{d}}_S$ and $\hat{\mathbf{d}}_T$

Line 312: added $\hat{\mathbf{d}}_T$

Line 324: changed "transient" to "multi-annual"

Line 409: added "multi-annual ice speeds"

**Comment 3.** Along the lines of future work, I would like the authors to add some brief discussion on the use of in-situ measurements to measure the propagation of waves. For example, can terrestrial radar or laser scanners be used to provide high-temporal-resolution measurements that can help further constrain glacier waves? This can be added to the end of Section 5.2.

We added the following sentences to (now) Section 5.3:

"A similar constraint may be obtained from terrestrial radar instruments that record velocity variations at timescales of minutes, allowing for high-resolution observations of dynamic responses to calving events or mélange collapse (e.g. Xie et al., 2019; Cassotto et al., 2019). In those situations, temporal basis functions and spatial correlations between basis functions can be used for dictionary construction and time-series inversions."

**Comment 4.** In a few parts of the discussion, it is stated that velocity and surface elevation are responding to changes in calving front position but this causality is not shown by the results of the paper. It is shown

definitively that variations initiate at the terminus and propagate upstream and that these variations are well-correlated with terminus motion. However, causality (in one direction or the other) is not shown by the analysis here. The language surrounding this discussion should be revisited and revised. Perhaps I have missed something and this causality can be inferred but, in that case, it needs to be made more explicit and clear in the discussion. Otherwise, the causality wording should be changed to discussing the correlation between terminus position and velocity.

While several prior studies have demonstrated causality between calving front position and velocity and elevation variations, we agree that our results as presented do not show causality on their own.

We modified the first paragraph in the Discussion to read:

"Decomposition of the time-dependent velocity and surface elevation fields into distinct temporal scales reveals a repeating pattern on Jakobshavn where velocity and surface elevation variations originate at the terminus. The coincidence between speedup and slowdown of the glacier with thinning and thickening, respectively, suggests a dynamic origin to the physical mechanism generating these variations. Prior studies have proposed that this mechanism is primarily characterized by a reduction of back stress at the terminus following a series of calving events, causing ice acceleration and increased driving stresses to propagate upstream which results in the observed high correlation between calving front position and velocity variations (e.g., Nick et al., 2009; Joughin et al., 2012; Bondzio et al., 2017)."

In the first sentence of Section 5.1.1, we removed "in response to changing calving front position".

Line 506: changed "perturbing forces" to "proposed perturbing forces".

In the first sentence of the Conclusion, we modified the following sentences (added text in bold):

"Over Jakobshavn Isbræ, this decomposition permitted a detailed investigation into the spatiotemporal characteristics of the evolving seasonal cycle of ice speedup and slowdown **which are shown to be highly correlated to seasonal terminus variations**. Analogously, longer-term changes in velocity were isolated and **also highly correlated with** longer-term terminus variations."

**Comment 5.** Finally, a minor comment that applies throughout the paper. There are a couple of places where the "southern bend" of the glacier is mentioned and I suggest adding something that indicates this region to all of the map-view figures.

We added an annotation for the southern bed in Figures 4 and 5 which show the phase delays for the short- and long-term velocity variations. However, we decided to omit the annotation for Figures 1 and 6 since the southern bend itself is not a central aspect of our results and is not closely related to the features shown on those maps.

**Minor comments:**

[line 50] What are sub-epoch velocity changes?

"Sub-epoch" is meant to imply temporal interpolation of the data between observation times by the estimated, continuous time series model. We changed the sentence to read "...while also allowing for interpolation of velocity changes between observation times throughout the glacier".

[line 137] This sentence can be removed.

We removed the sentence.

[Fig 1] State in the caption that the map coordinates are polar stereographic north (EPSG:3413).

Added to caption.

[Fig 1] I suggest replacing the manually drawn white lines in panel A with either the calculated contours of maximum shear strain rate or with the contour where the bed is at sea level. This would be a more accurate depiction of the trough and the main trunk of the glacier.

We have replaced the drawn white lines with a zero-meter-elevation contour of the bed and captions.

[Fig 1] The sentence "Mean velocities are added to time series for visual clarity" does not make sense to me. How are the mean velocities depicted in the plots? And what are these means (spatial? temporal?)?

We modified this sentence to read: "For each time-series, mean velocities for 2009 – 2019 have been added as offsets for visual clarity."

[Fig 1] Clarify how the data is detrended in C and D. For example, something like: "white dots indicate (B) observed speeds, (C) observed speeds detrended using seasonal splines, and (D) observed speeds detrended using seasonal and transient splines."

We added the following sentences to the caption: "The detrended short-term observations in C are the observed speeds minus the estimated integrated B-splines. The detrended long-term observations in D are the observed speeds minus the estimated seasonal B-splines."

[Fig 2] The approximate solid black lines drawn are very helpful in illustrating wave propagation and it is clear from the differences in panels A and B that phase speeds of seasonal signals are much different from multi-year signals. However, I would like to see calculated contours drawn on each panel. These could be the zero contours or any other arbitrary value and they can be displayed in grey, with the approximate lines in darker black for illustrative purposes.

We added zero-velocity-contours for the leading edges of the events (summer speedups for the seasonal velocity variations and 2012 speedup and 2017 slowdown for the multi-annual variations). We felt this approach was a good compromise between showcasing the data and limiting visual clutter.

[line 269] I am confused by the phrase "long-term signals removed." Is removing the long-term signal the same as combining the transient and seasonal signals? In other words: $d_L = d - d_T - d_S$, where $d_L$ is the long-term signal shown in Fig. 1C $d - d_L = d_T + d_S$ If this is the case, I suggest rewording this from "the velocity data from 2011 to 2018 at each pixel with the estimated long-term signals removed, $d_S$" to "the combined seasonal and transient modeled signal, $d_T + d_S$"

"Removing the long-term signal" means removing the transient signals from the full model fit. In this paper, long-term == multi-annual == transient, as described in the modified Section 2 and shown in Fig. 1D. Therefore, we are using Equation 7 to model the time-series in Fig. 1C (as stated in the text). We believe that our modifications to the main comments (1) and (2) provide sufficient clarity for the reader to follow which time-series are being discussed.

[line 300] I would replace "classical" with "time-series".

Done.

[line 314] This is the only place in the paper where Fig 3C is referenced and I think it is completely OK to hypothesize about the connection between phase velocity and thickness/bed but, because a figure is provided, I would suggest expanding on this a bit. Please add a sentence that explicitly states the hypothesis about the relationship between these two variables (e.g., higher/lower velocity in thicker/thinner ice).

We added the following sentence: "For glaciers where ice flow is dominated by basal sliding, phase velocity is expected to scale with the square root of ice thickness and basal shear traction (Rosier et al., 2014), which

is roughly consistent with the increase in phase velocity and ice thickness around 8 km upstream, although more work is needed to establish concrete connections."

[lines 332-334] This sentence is accurate but the "while" clause does not make sense to me. I am reading it as "along the trunk there is lowering, while on the slower ice, there is lowering." Please clarify. Perhaps this sentence is meant to say that there is a confined region of high thinning along the trunk and near the front, while on the slower ice there is still thinning but lower magnitude.

We changed the sentence to read: "Within the main trunk of the glacier, we observe a clear association between the 2012 speedup and lowering of the ice surface due to dynamic thinning, whereas on the slower ice, thinning is more diffuse and occurs at a lower rate."

[lines 334-336] This sentence makes two claims without providing evidence. First, that the slower ice was thinning before the observation period. Second, that high melt started in 2009. Both of these must be backed up with either a figure or a citation.

After more discussion, we decided to remove references to high melt prior to the observation period since thinning of the inland areas in the 2000s for Jakobshavn is more likely a dynamic response to the speedup in the main trunk in 2004 (following disintegration of the ice tongue). We have modified this sentence to read:

"A comparison of time series for points on and off the glacier (Figure S1) suggest that much of the ice in the surrounding areas has been lowering since before the observation period. In these areas, thinning has been attributed to inland diffusion of steepening surface slopes following speedup and thinning of the fast-flowing trunk in 2004 in response to disintegration of the ice tongue (Krabill et al., 2004; Joughin et al., 2008)."

[line 360] I would find it helpful to distinguish the results presented in this paper from earlier work here. Adding a clause to this sentence such as, "Consistent with earlier work ..., but at a higher-temporal resolution, we observe ..." or "Consistent with earlier work ..., but using our novel method that is better able to isolate seasonal signals, we observe ..." [lines 360-361] This paragraph and the corresponding figure describes the relation- ships between (1) seasonal terminus positions and seasonal velocity variations and (2) long-term terminus positions and long-term velocity variations. Thus, I suggest re- wording this sentence from "we observe a strong correlation between the seasonal variations in ice velocity and the year to year variations of the front" to "we observe strong correlations between variations in ice velocity and variations of the front at both the seasonal and long-term time scales"

We modified the sentence to read:

Consistent with earlier work (e.g. Joughin et al., 2012; Cassotto et al., 2019) but using our method to decompose velocity time-series into short- and long-term variations, we observe strong correlations between variations in ice velocity and variations of the front at both seasonal and long-term time scales.

[Fig 7] I suggest using a sequential colorscale, rather than a divergent one, to represent different years. The current colorscale makes it impossible to distinguish 2009 from 2018.

We chose the cyclical color scale to emphasize the different seasonal behaviors between the two different clusters of years, rather than emphasizing any one year specifically. However, we updated the figure to use dashed lines for 2009/2010 to make it easier for the reader to distinguish between individual years (as suggested by Reviewer 2).

[Fig 7] In panel D, in addition to coloring the points according to year, distinguish the two groups separated by terminus position using different symbols (e.g., circles and squares).

We have modified the figure and caption accordingly (diamonds and circles).

[Fig 7] In panel D, it is not clear to me how the seasonal velocity variation quantity is calculated. Please add this to the text or the caption.

Here, we are simply using the reconstructed seasonal velocities, $\hat{\mathbf{d}}_S$. We have modified the sentence introducing panel D as: "D) Correlation between terminus position and seasonal velocity (i.e., $\hat{\mathbf{d}}_S$) at the same location as in (A) and (B)".

[lines 392-395] Strictly speaking, the results do not show that velocity and surface elevation variations are changing in response to changes in the calving front position. They are certainly correlated but causality one way or another has not been shown here. I suggest rewording this.

Please see response to main comment 4.

[line 410] I would add the word "transient": "... as well as the transient response ..."

Done.

[lines 410-411] This sentence is a bit confusing to me. What is meant by quantifying "wave propagation to phase velocities and attenuation length scales?" Does this mean quantifying the relationship between wave propagation distance(?) to phase velocities and attenuation length scales? Something seems to be missing here.

This sentence was meant to convey that we cannot fully quantify wave propagation behavior (e.g., dispersion relations, modification of waveform shape with upstream distance, etc.) with the data resolution we have. We can only quantify the scalar quantities of phase velocity and attenuation length scale. We changed the sentence and the following sentence to read:

"At the moment, data sparsity only allows for quantification of phase velocities and attenuation length scales for describing overall wave propagation behavior. As more data become available, the time series methods outlined above should allow for observations of waveforms manifest in surface elevation fields and broader and more refined constraints on the functional form of dispersion relations (the relationship between frequency and wavelength) for individual glaciers."

[lines 412-413] Please reword to be more explicit about what is meant by "broader and more refined constraints". Does broader mean for more glaciers or at more frequencies? Does more refined mean smaller uncertainties? If so, state this explicitly.

We modified the second half of that paragraph to read (added text in bold):

" As more data become available, the time series methods outlined above should allow for observations of waveforms manifest in surface elevation fields **in addition to constraints on dispersion relations (the relationship between frequency and wavelength) on individual glaciers that cover a broader range of frequencies with finer resolution in the frequency domain**. Realizing this potential for remote sensing time series is important because the characteristics of wave propagation, specifically the dispersion relation as defined for a wide range of frequencies, are intrinsic properties of dynamical systems, if we define the system in this case such that it includes the glacier and boundary conditions. **As such, time-dependent velocity and elevation data for glaciers characterized by a wide range of sliding speeds and geometries can be used to determine the relative contributions of forcing frequency, ice thickness, glacier width, and basal traction on measured phase velocities attenuation lengthscales, thereby providing a method for inferring relevant mechanical and rheological parameters.**"

[line 489] Can anything more be added about the kinds of waves that are observed on Rutford? Did previous work categorize what kind of waves those are? If so, I would add that here to enhance the contrast between the kinematic waves on Jakobshavn and the other type of wave on Rutford.

We are currently working on a theoretical framework for relating wave propagation at sub-annual timescales to intrinsic physical properties of laterally-confined glaciers and ice streams. To our knowledge, no previous work exists that fully characterizes wave and dispersion behavior at these timescales. Nevertheless, the mechanisms proposed for driving velocity variations at Rutford all share the feature of dynamic redistribution of longitudinal stresses (and possibly combined with basal drag reduction through subglacial hydrology).

We re-organized the paragraph a bit and added a sentence that explains the above comment in more detail. The relevant portions are as follows (added text in bold):

"On Rutford — with a mean flow speed near the grounding line of approximately 375 m/year — velocity variations driven by ocean tides propagate upstream with a phase velocity of approximately 24 km/day for the first 40 km upstream of the grounding line, and then at a faster rate of 34.3 km/day further upstream. Thus, observed waves on Rutford propagate two orders of magnitude faster than those we observe on Jakobshavn. The attenuation length scales are also markedly different: approximately 45 km on Rutford versus 7 km (seasonal) and 14 km (multi-annual) on Jakobshavn. Several observational and modeling studies have suggested that at Rutford, downstream variations in buttressing stresses over the ice shelf, grounding line position, and/or water pressure at the bed (Gudmundsson, 2006, 2007; Rosier et al., 2014, 2015; Minchew et al., 2017; Robel et al., 2017; Rosier and Gudmundsson, 2020) drive the variations in flow speed over the tidal cycle. The marked differences in forcing frequencies and propagation speeds and distances between Rutford and Jakobshavn therefore suggest fundamental differences in wave types and forcing mechanisms. **Indeed, while previous work cited above suggests that waves on Rutford are influenced by the viscoelastic properties of ice expected at fortnightly periods, the much longer periods of variability on Jakobshavn render elasticity of glacier ice negligible and thus unlikely to contribute in any meaningful way to wave propagation. We further discuss the distinctions between wave types below.**"

[line 597] "IceSat-2" should be changed to "ICESat-2"

Done.

[line 631] Please add a sentence describing, briefly, the caveats to the conclusion that the observed waves are kinematic in nature. These caveats are very nicely discussed in detail in Section 5.1 and I think they need to be summarized in the conclusion.

We added the following sentence: "However, the dispersive nature and higher phase velocities of the observed waves relative to previously proposed kinematic waves necessitates further investigation into their physical drivers and the overall dynamic response of glaciers to stress and mass perturbations."

[Data availability] Please add the DOI for the OMG DEMs (`https://doi.org/10.5067/OMGEV-GLNA1`)

Done.

**3    Response to Reviewer 2**

**Comment 1.** My main criticism is that it was difficult to follow why and how different time periods were used for various analysis throughout the paper. I would like to see either more coherence in selected time periods used, or more description up front in the introduction/motivation to explain why various subsets of the time period are used at different points throughout the text. For example, the abstract and introduction

refer to a 2009- 2019 decadal study, shown in completeness Figure 1. However, Figure 2 then only shows 2011-2018, and subsequent analysis average similarly segmented time periods. As another example, mean time of peak seasonal velocity and phase velocities seem to be computed using only 2011-2018 velocity data.

As a general note, our analysis strategy was to focus on the years where the seasonal velocity variations were similar with regards to amplitude and upstream propagation characteristics. The years 2009-2010 were somewhat anomalous in this regard (especially 2010), mainly in terms of timing of the seasonal speedups. Therefore, for computing and presenting the seasonal phase delay (e.g., Figure 3), it was important to not introduce bias into the phase estimation from the years where the seasonal cycle was shifted within the year. In Section 4.1, we added the following sentences to motivate these decisions:

Line 277: "...occurring around mid-September. The exception to this timing is the 2010 speedup which starts earlier in the year and may have been driven by a combination of warmer air temperatures and cooler ocean temperatures influencing mélange rigidity during the course of the seasonal cycle (Joughin et al., 2020)."

Line 294: "Note that the years 2009-2010 are excluded in order to avoid introducing biases into the phase estimation from differences in onset times of summer speedups."

However, we do see the value in visualizing the entire time span of the data in order to clearly see the change in short-term velocity variations after 2012. We have modified Figure 2 to include the entire time span (please also refer to our response to Reviewer 1 where we have added zero-velocity contours to the images). From the updated Figure 2, we can now see that while the summer speedup in 2010 has an earlier onset time than the subsequent years, the phase velocities for 2009-2010 are quite similar to the other years.

**Comment 2.** Reference point used for correlation analysis: Why is a point 1.4 km upstream of the pinning point used specifically for comparison to velocity? I see that you found the highest temporal coherence between maximum retreat and maximum seasonal velocity at this location, but it would be helpful to have more information on how this coherence was derived, and why then it serves as an ideal reference point.

Please also include a map view of the reference point and pinning point on the map. I found it hard to follow why sometimes the point 1.4 km upstream of 2017 front was used, versus the pinning point, as reference in the figures. What information is lost if, as a suggestion, the pinning point is used as the single point of reference throughout the text?

The main reason the pinning point is not used as the reference point is data availability: velocity estimates are generally not available downstream of the calving front (where they do exist, the velocity estimates are of the melange and not the grounded ice). Therefore, we chose a reference point where velocity data were available for the entire time span of the data. Strictly speaking, full data availability is not required since we can interpolate through temporal data gaps with the B-splines, but we would risk over-smoothing the seasonal velocity variations, particularly in the summer months where the front retreats past the pinning point. Overall, the reference point was not chosen to maximize coherence between front position and velocity; it was simply chosen to remain close to the calving front while ensuring as much velocity data are available for our analysis time period. In fact, due to the fast wave speeds of seasonal variations in the first 5-10 km upstream of the 2017 front (Figure 3), any point within that region would show very similar coherence with the front timeseries. Towards the end of Section 4.4, we also cite a recent study that compared velocity to front position for a moving point: "...comparison of front position with velocities at a moving point 1-km upstream of the front still show lower correlation for the years 2009 – 2010 (Joughin et al., 2020)...".

To de-emphasize the choice of "1.4 km upstream", we have modified the text to read (Line 383):

"The timing of maximum retreat for a given year is closely associated with the timing of peak seasonal ice velocity within a few km of the front position. Here, we choose a point approximately 1.4 km upstream of the 2017 terminus in order to maximize data availability close to front position for all years."

We have added markers in Figure 4 to indicate the location of the reference and pinning points. We have also added a supplemental figure (Figure S2) showing the location of the pinning point relative to contours of the bed topography.

**Comment 3.** Multiyear variations in surface elevation: It would be helpful to have more text describing the motivation for selecting the particular time epochs shown in Figure 6. Each of the 4 panels represent elevation changes over time intervals of varying lengths, from ?1.5 years to ?2.5 years. The selected intervals also exclude July 2015-December 2015. There may be a reason for this, but without more context it seems too arbitrary.

Our original motivation for selecting those time periods was to emphasize distinct velocity and elevation change patterns corresponding to dynamical events (e.g., 2012 speedup and 2017 slowdown). However, we agree that this rationale can be opaque to readers, so we remade Figure 6 to show velocity and elevation differences for uniform time periods of 2.2 years (see Figure S1 to see how the time periods brackets the aforementioned dynamical events). The derived conclusions are unchanged, and the maps successfully emphasize our original intended features.

**Comment 4.** Discussion: I would like to see the discussion expanded to include considerations/limitations of this framework when applied to other glacier sites outside of Jakobshavn Isbræ (for example, in areas with notably lower SNR than Jakobshavn). Do you anticipate reduced SNR would limit the feasibility of this technique (and how may opting to enforce spatial coherency impact interpretations of phase velocity, propagation delay, etc).

We have added a section in the Discussion (Section 5.2) that addresses the applicability of these methods to other study areas and the main challenges we foresee. We include the text here for completeness:

"5.2 Applicability to other study areas

The GIMP velocity data over Jakobshavn Isbræ has high SNR for both the short- and long-term variations, which facilitates reconstruction of the spatiotemporal evolution of the traveling waves discussed in this work. Additionally, the dense temporal sampling relative to the signals of interest avoids potential issues related to oversmoothing of short-term velocity variations. However, many other glaciers and ice streams in Greenland and Antarctica will not have the same level of data coverage as Jakobshavn, which may limit the recovery of similar dynamical signals. Data coverage in this context is specified by temporal sampling and spatial continuity of velocity data where the former is likely to be the primary limiting factor for time-series analysis. For example, velocity data derived primarily from optical platforms are generally restricted to the summer months where cloud and snow cover effects are minimized. This asymmetry in coverage for a given year will alias reconstruction of seasonal velocity cycles, which would likely cause artifacts when attempting to quantify wave properties like phase velocity. We estimate that velocity data provided at monthly intervals constitute the lower bound for temporal resolution in order to quantify wave behavior at sub-annual timescales using the methods presented here. Of course, higher phase velocities for certain classes of dynamical signals may necessitate remote sensing data with finer temporal resolution (Minchew et al., 2017).

Spatial resolution and spatial data gaps can also limit characterization of wave behavior and other changes in ice flow. For example, regions near glacier termini will undergo periods of missing data associated with termini retreat where velocity data cannot be obtained over open water. The temporal interpolation properties of B-splines can mitigate these effects to some degree, but study areas with more persistent spatial gaps will

likely benefit from incorporation of spatial coherency, which enforces that neighboring grid points share similar temporal behavior. However, data that require stronger levels of spatial coherency may also result in reconstructed signals that are oversmoothed, which would bias phase velocities and decay lengthscales to lower and higher values, respectively. In these situations, it would be beneficial to incorporate independent data sources like GPS time-series to provide additional validation data for 'tuning' the time-series analysis parameters. Overall, we expect that current and future remote sensing platforms will provide high-quality data similar to the GIMP data over Jakobshavn Isbræ, and we discuss those implications next."

Figure 2: Not a critique, but comment: This is a great figure that illustrates a lot of information in a concise way, clearly showing interannual variations in amplitude and inland propagation of signals from the front. The figure caption also included an excellent description of how phase velocity was extracted from the tangent angle.

Thank you! Please note that we added zero-velocity contours to Figure 2 to address comments by Reviewer 1 (in addition to extending the visualized time period).

**Minor comments:**

Figure 3a: I suggest scaling the y-axes such that a range of the same magnitude is shown for both. This would allow the reader to quickly compare relative changes in slope with distance between mean velocity without amplitude.

We experimented with a common y-axis for Figure 3A, but we found that it was more difficult to visualize the differences in decay lengthscales between the seasonal and multi-annual signals. As a compromise, we scaled the mean velocity axis to be twice that of the amplitude axis to make it easier for the reader to perform a conversion between the two. We updated the caption by adding: "(note the 2x scaling factor for the mean velocity axis)".

Figure 4: why are data from 2016 excluded from either group?

2016 was somewhat of an anomalous year for the seasonal cycle because the calving front had not sufficiently advanced during the winter months (Figure 7C), leading to a velocity at the beginning of the summer season that was higher than the other years from 2011 - 2018. We discuss this in the text around Line 383, and added a clarifying sentence in the caption for Figure 4.

Figure 5: Please add a note to caption to remind reader that red lines delineate winter 2017 reference calving position.

Done.

Figure 7a and c: It is very difficult to differentiate between years 2009/2019 and 2017/2018. If keeping the same color scale is preferred, I suggest making 2009/2010 dashed rather than solid lines to make years more distinct.

We have updated the figure to use dashed lines for 2009/2010. We prefer this cyclical colormap in order to emphasize differences in behavior between the two clusters of years, but we agree that the dashed line helps the reader better distinguish individual years within the clusters.

Figure 7d and correlation analysis: Are the velocity values shown here (and used for correlation analysis) taken from the continuous fitted time series? If so, what is the sampling frequency from these curves (every week, every month?) Are the extracted velocity values uniformly spaced in time?

For all reconstructed (continuous) time series of velocity and elevation, the sample spacing is approximately 4 days, spaced uniformly in time. We added a sentence to the beginning of Section 4 to state this.

Line 334: "A comparison of time series for points on and off the glacier suggest that much of the ice in the surrounding areas has been lowering since before the observation period, which is associated with a period of exceptionally high surface melt starting around 2009." Can you include a citation for this?

After more discussion, we decided to remove references to high melt prior to the observation period since thinning of the inland areas in the 2000s for Jakobshavn is more likely a dynamic response to the speedup in the main trunk in 2004 (following disintegration of the ice tongue). We have modified this sentence to read:

"A comparison of time series for points on and off the glacier (Figure S1) suggest that much of the ice in the surrounding areas has been lowering since before the observation period. In these areas, thinning has been attributed to inland diffusion of steepening surface slopes following speedup and thinning of the fast-flowing trunk in 2004 in response to disintegration of the ice tongue (Krabill et al., 2004; Joughin et al., 2008)."

Line 387: "After the disintegration of the ice tongue between 1998 and 2004, the front rapidly retreated about 4 km over the period from 2004 to 2011." Please include a citation for this, as front analysis used in this study did not start until 2009.

We actually use calving front data from the Greenland Ice Sheet Climate Change Initiative (CCI) from 2000 to 2016 (see Figure 7 where we compare velocity and front position starting in 2004). In the text in Section 3.3, we originally had a typographic error by stating the CCI data spanned from 2009 to 2016. We have corrected this. Additionally, we added a citation for the ice tongue disintegration between 1998 and 2004 (Joughin et al., 2004).

[revised manuscript text omitted]
})$. Therefore, each reference function is evaluated over the entire timespan of the time series and placed into the columns of $\mathbf{G}$. An important advantage of using a linear model  is the ability to evaluate the  reference functions (and thus construct $\mathbf{G}$) at any arbitrary time, which provides a natural way to assimilate time series with missing or irregularly spaced data. Additionally, linear models facilitate the use of powerful and efficient linear regression inverse methods to solve for the coefficients in $\mathbf{m}$ (Tarantola, 2005).

The dictionary $\mathbf{G}$ can contain any combination of functions that collectively capture the observable temporal variations. Thus, the inverse problem for $\mathbf{m}$ is often ill-posed because the dictionary $\mathbf{G}$ can be overcomplete, with many more reference functions (columns) than observations (rows). Therefore, we use regularized least squares to obtain an estimate $\hat{\mathbf{m}}$ that minimizes a cost function containing the data residual and regularization terms, such that (Riel et al., 2014, 2018)

[revised manuscript text omitted]

**4.1   Seasonal Variations in Surface Velocity**

The reconstructed velocity time series demonstrates the ability of our flexible method to smoothly interpolate the velocity data in time in a manner that preserves the seasonal variations. In particular, the use of temporally coherent B-splines to model

[Figure]

**Figure 1.** Sermeq Kujalleq (Jakobshavn Isbræ) mean velocity field and select velocity time-series. A) Mean velocity between 2009 and 2019 in polar stereographic north (EPSG:3413) coordinates. The background image is a Landsat 8 image acquired on August 2017. White lines correspond to the bed topography (BedMachine V3) at sea level, and the red line indicates the winter 2017 terminus position (for all subsequent map figures). White triangles indicate the points P1, P2, and P3 from which data shown in panels B–D are taken. Inset shows (with red arrow) the approximate study area within Greenland with mean velocities from Joughin et al. (2011). B) Time-dependent speed at points P1, P2, and P3. C) Short-term velocity time-series showing predominantly seasonal variations. For each time-series, mean velocities for 2009 – 2019 have been added as offsets for visual clarity. D) Long-term velocity time series showing the 2012 speedup and 2017 slowdown. In panels B–D, solid lines show our model results while dots indicate (B) observed speeds or (C and D) detrended observationsThe detrended short-term observations in C are the observed speeds minus the estimated integrated B-splines. The detrended long-term observations in D are the observed speeds minus the estimated seasonal B-splines.

seasonal variations allows for reconstruction of several summer speedup events where data happen to be more sparse for certain
250   years (Figure 1). By applying Eq. 5 to decompose the velocity magnitude time series into seasonal and transient components ($\hat{\mathbf{d}}_S$ and $\hat{\mathbf{d}}_T$, respectively), we show that short-term velocity variations on Jakobshavn are dominated by the seasonal cycle of summer speedup and winter slowdown. In this section, we focus on these seasonal variations, leaving discussion of the multi-annual transient variations for the next section.

[Figure]

**Figure 2.** Temporal evolution of glacier flow variations along a centerline segment for decomposed seasonal (A) and multi-year (B) signals. The centerline trace is shown in map view in Figure 4, and distance values here are measured upstream along the centerline from the winter 2017 terminus position.  Thin black lines  correspond to contours of zero-velocity variation at the initiation of each signal (summer speedups for (A) and 2012 speedup and 2017 slowdown for (B)). Solid grey lines approximately follow the zero-velocity contours and indicate the leading edges of propagating  velocity variations.  Vertical  grey dashed lines indicate onset times for the propagating wave initiating at the terminus and are equivalent to the propagation path for a wave with infinite propagation speed. The tangent of the angle between solid and dashed grey lines is the phase velocity averaged over the observable propagation distance. The marked difference in phase velocities between the seasonal and multi-year signals indicates (frequency) dispersion.

The flexibility of the B-spline representation for the seasonal time series allows us to quantify the change in the amplitude
255   of summer speedup from year-to-year and at each point on the glacier. In Figures 1 and 2 we show these variations in two different views to aid in interpretation of the results. Figure 1A,B represents a classical view of spatiotemporal variations in surface velocity, with a map of secular velocity (Figure 1A) and time series of select points on the glacier (Figure 1B). Figure 1C shows, for the same points on the glacier, the seasonal variations, which are the total signal shown in Figure 1B less the inferred multi-year trends discussed in the next section. In Figure 2, we present a space-time plot for the (A) seasonal
260   and (B) multi-year variations along the centerline transect shown in Figure 4A. This representation allows for an intuitive visualization of spatiotemporal variations in the surface velocity fields and, most relevant for this study, the propagation of velocity variations through the glacier in time. Our analysis focuses on this propagation by treating velocity variations as traveling waves with quantifiable attenuation and propagation rates. To aid in this discussion, we have provided a visual representation of the upstream propagation rate in Figure 2 using the solid and dashed black lines. The angle between the

[revised manuscript text omitted]

**4.2 Multi-Year Variations in Surface Velocity**

After isolating the long-term signals from the short-term seasonal signals  (i.e., $\hat{\mathbf{d}}_T$), we can observe clear variations in multi-annual amplitudes at different points along the glacier (Figures 1D and 2B). The temporal density of the velocity time series allows us to quantify spatial variations in the amplitude and timing of the positive and negative multi-annual trends, much like we did with seasonal velocity variations in the previous sections. We observe two events in the data: a speedup that begins in 2012 and a slowdown that begins in late 2016 near the terminus, which we refer to as the 2017 slowdown. We present the results for multi-annual variations using the same general structure as for the seasonal, with the  time-series view shown in Figure 1, the space-time diagram in Figure 2, along a centerline transect in Figure 3, and map view of the amplitudes and phase values in Figure 5.

The spatial pattern of the amplitudes of multi-annual velocity variations is remarkably consistent between the two observed events, with the highest amplitudes at the terminus and an exponential decay with distance upstream (Figures 3A and 5A–B). Notably, the velocity variations induced by these events have an attenuation ($e$-folding) length scale of approximately

[Figure]

**Figure 4.** Seasonal variations in flow speed and timing of peak velocities. A) Mean seasonal velocity amplitude for the years 2009 – 2011 and 2017 – 2018 (years not associated with the increased velocities between 2012 and 2015). Triangle markers indicate comparison points used in Figure 7: pinning point (PP) where the bed topography locally narrows; and reference point 1.4 km upstream from the 2017 front position ($P_{1.4}$). B) Mean seasonal velocity magnitude variation for the years 2012 –  2015 (2016 excluded due to higher background velocities at the start of the summer). C) Mean day-of-year of peak seasonal velocity (i.e., seasonal phase) for entire observation period. SB indicates the southern bend referred to in the text. D) Seasonal phase uncertainty ($1\sigma$). Seasonal amplitudes are measured as the difference between the summer high and winter low velocities in the short-term time series as shown in Figure 1C. The highest amplitudes occur at the terminus and decay exponentially upstream.

$14.1 \pm 0.3$ km, which is about twice the attenuation length for seasonal variations (Figure 3A). As a result, we are able to observe multi-annual velocity variations farther upstream than the seasonal timescale velocity variations (Figure 2B).

From the phase delay of the  multi-annual signals, we can see that both the 2012 speedup and 2017 slowdown signals originate at the terminus, propagate rapidly along the first 5 km of the glacier, slow down through the southern bend between 5 - 8 km, and propagate upstream from 8 - 20 km at a generally consistent phase velocity (Figures 3 and 5C–D). Beyond 20 km upstream, the amplitudes for the velocity variations become too low to reliably estimate the timing of arrival of the transient signals (Figure 5D). The phase velocity between 8 and 20 km upstream is approximately $231 \pm 16$ m/day ($84 \pm 6$ km/yr), which is a little more than half of the phase velocity for the seasonal signal and roughly seven times the mean flow

335

speed near the terminus.  For glaciers where ice flow is dominated by basal sliding, phase velocity is expected to scale with the square root of ice thickness and  basal shear traction (Rosier et al., 2014), which is roughly consistent with the increase in phase velocity and ice thickness around 8 km upstream, although more work is needed to establish concrete connections. As with the observed seasonal variations, the phase velocity in the first 5 km upstream of the terminus is at the limit of the temporal resolution of the data with a lower bound on the phase velocity of at least 500 m/day (182.5 km/yr), or approximately 18 times the mean glacier flow speed in this region. While the slowdown in wave propagation in the southern bend is coincident with a local high in the bed topography, more work is needed to evaluate whether the topographic effect is the dominant control on wave propagation. The apparent slowdown may also be an artifact of numerical errors caused by tracking of peak acceleration/deceleration rather than a multi-year average of sinusoidal phase as for the seasonal signal. Nevertheless, the consistency in the spatial distribution of peak timing and amplitude reinforces the notion that a common physical mechanism is responsible for multi-year and seasonal velocity variations.

[Figure]

**Figure 5.** Velocity variation amplitude and phase delay maps for 2012 transient speedup (A and C) and 2017 transient slowdown (B and D). In C, SB indicates the southern bend referred to in the text. Red line indicates winter 2017 terminus position. In addition to the spatial distribution of phase delay and amplitude being consistent for the two events, these transient phase delays show strong similarities with the seasonal phase delay. However, the transient amplitudes show farther upstream propagation than the seasonal amplitudes.

**4.3 Multi-Year Variations in Surface Elevation**

Ice surface elevation varies in response to changes in snow accumulation and melt (the sum of which constitutes the surface mass balance; SMB), firn compaction (Herron and Langway, 1980; Huss, 2013; Meyer et al., 2019a), and dynamic thinning (thickening) in response to increases (decreases) in the flux divergence of the ice. The interplay between observed elevation and velocity changes at different temporal and spatial scales can thus yield insight into the mechanisms driving longer-term elevation and velocity changes.

For this work, the temporal sampling of the available elevation data (ArcticDEM and OMG DEMs) permits only the comparison of longer-term variations in velocity and elevation. Thus, we compare the long-term velocity and elevation changes for four  successive time periods of length 2.2 years: 1)  June 2010 to September 2012; 2)  September 2012 - November 2014; 3)  November 2014 -  January 2017; and 4)  January 2017  to March 2019 (Figure 6).  Within the main trunk of the glacier, we observe a clear association between the 2012 speedup and lowering of the ice surface due to dynamic thinning, whereas on the slower ice, thinning is more diffuse and occurs at a lower rate. A comparison of time series for points on and off the glacier (Figure S1) suggest that much of the ice in the surrounding areas has been lowering since before the observation period. In these areas, thinning has been attributed to inland diffusion of steepening surface slopes following speedup and thinning of the fast-flowing trunk in 2004 in response to disintegration of the ice tongue (Krabill et al., 2004; Joughin et al., 2008). The widespread lowering of the ice surface persists even during the slight deceleration of velocities over the glacier from 2012 to 2015 (Figure 6C-D). During these years, the ice surface has likely adjusted to the initial speedup in 2012, leading to a reduction in driving stress and velocities. The 2017 slowdown is coincident with thickening of the ice on the main trunk of the glacier while the inland ice continues thinning (Joughin et al., 2018; Khazendar et al., 2019; Joughin et al., 2020). From 2017 onwards, we observe more widespread thickening of the ice as the glacier velocities continued to decrease. Despite the recent thickening trend, elevation values are still measurably lower than they were in 2010, particularly for the ice outside of the main trunk of the glacier (Figure S1).

**4.4 Calving Front Forcing**

The position of the calving front has been reported to be the dominant factor influencing ice dynamics for Jakobshavn Isbræ over observable, particularly seasonal, timescales (Nick et al., 2009; Joughin et al., 2012; Bondzio et al., 2017). The position of the calving front is heavily influenced by bed topography (Xu et al., 2013; Morlighem et al., 2016), local ocean temperatures that influence subaqueous melting of the terminus (Holland et al., 2008; Khazendar et al., 2019), and changes in melange rigidity (Joughin et al., 2008; Cassotto et al., 2015; Robel, 2017; Xie et al., 2019; Joughin et al., 2020). Changes in the calving front driven by iceberg calving can reduce back stresses upstream of the new calving front, allowing ice near the terminus to accelerate and subsequently thin. Local thinning of the ice steepens the surface slope, thereby increasing gravitational driving stress and causing further acceleration and thinning (Joughin et al., 2012). For glaciers where the calving front is located

[Figure]

**Figure 6.** Long-term velocity and ice surface elevation changes for  successive time periods of 2.2 years. (A-B)  June 2010 to  September 2012: The glacier is accelerating (positive velocity anomaly) and thinning (negative elevation anomaly), with higher rates of thinning within the glacier. (C-D)  September 2012 to November 2014: Velocities show slight deceleration while the ice surface lowers over a wider area, indicating persistent surface melt. (E-F)  November 2014 to  January 2017: This time period contains the  initiation of the 2017 slowdown, which is associated with ice thickening in the main trunk of the glacier. Note the continuing surface melt signal indicated by lowering distal to the glacier. (G-H)  January 2017 to March 2019:  Slight decrease in glacier flow speed and more widespread increase of ice surface elevations due to positive SMB can be seen.

on a retrograde bed, thinning-induced retreat of the terminus corresponds to increasing terminus ice thickness, which further
385   increases the driving stress (Joughin et al., 2020). The redistribution of mass during dynamic thinning propagates upstream as traveling waves, with phase velocities and attenuation length scales that we constrain using the methodology presented in this study. The thinning of the ice surface throughout the glacier is hypothesized to cause a further, indirect enhancement of

velocities by causing a reduction in overburden pressure (weight of the ice column per unit area), which influences the response of the glacier to processes such as basal lubrication by drainage of surface meltwater in the summer melt season. While the role of basal lubrication has been shown to affect the seasonal cycle on certain glaciers (e.g. Howat et al., 2010; Bevan et al., 2015), the magnitude and timescale of the influence of basal lubrication on the velocities in Jakobshavn remains unresolved.

Consistent with earlier work (e.g. Joughin et al., 2012; Cassotto et al., 2019) but using our method to decompose velocity time-series into short- and long-term variations, we observe  strong correlations between variations in ice velocity and  variations of the front  at both seasonal and long-term time scales (Figure 7). The timing of maximum retreat for a given year is closely associated with the timing of peak seasonal ice velocity  within a few km of the front position. Here, we choose a point approximately 1.4 km upstream of the 2017 terminus  in order to maximize data availability close to front position for all years. For the four years associated with increased seasonal velocity amplitudes (2012 – 2015) during the summer, the calving front retreats past  a narrow section in the bed referenced by (Cassotto et al., 2019) ( defined as our reference position for the terminus position time series and shown in Figure S2) into the wider and deeper basin (values > 0 in Figure 7C), which supports previous hypotheses that even subtle bed constrictions in the fjord can lead to large increases in ice velocity in response to terminus retreat when ice elevations are near flotation heights (Cassotto et al., 2019). In 2016, the calving front also retreats past the same pinning point, but the seasonal velocities do not reach the same peak as the previous four years. In fact, in 2016, the calving front starts the summer melt season in a more retreated position, which is a consequence of the front not sufficiently advancing in 2015 and prematurely retreating in December of that year. Thus, while the seasonal amplitude in 2016 is less than the previous four years, the absolute velocities are still high (Figure 1C). For the years 2009 – 2011 and 2017 – 2018, the calving front does not retreat past the pinning point, which results in seasonal velocity variations with markedly smaller amplitudes.

Following Lemos et al. (2018), we compute the correlation between the measured calving front position (relative to the reference pinning point) and short-term velocities for the same point 1.4 km upstream of the 2017 terminus (Figure 7D). The change in the velocity response to front variations between 2012 and 2015 can be clearly seen as a distinct cluster as compared to the other years in the observation period. For these four years, a linear regression yields a coefficient of determination ($R^2$) of 0.71 with seasonal velocities scaled by 2.2 km/year per km of front retreat. For the other years, the regression results in an $R^2$ of 0.48 with a scale factor of 1.2 km/year per km of front retreat. Thus, from 2012 to 2015, the velocity response is more strongly correlated with front position with larger peak-to-peak variations than the other years, presumably due to the retreat of the front past the reference pinning point described by Cassotto et al. (2019). The higher level of correlation for the years 2012 – 2015 is likely driven in part by the proximity of our point of comparison (1.4 km upstream of the 2017 terminus) to the retreated front position for those years. However, comparison of front position with velocities at a moving point 1-km upstream of the front still show lower correlation for the years 2009 – 2010 (Joughin et al., 2020), which underscores the importance of bed topography on the response of ice flow to front position.

On timescales longer than a year, calving front positions and  multi-annual ice speeds also co-vary, but the relationship is more non-linear than on seasonal timescales (Figure 7B and D). After the disintegration of the ice tongue between

1998 and 2004 (Joughin et al., 2004), the front rapidly retreated about 4 km over the period from 2004 to 2011. During this time, the ice speed near the 2017 terminus increased about 1.5 km/yr, half of the roughly 3 km/yr increase associated with the

425    2012 speedup. The 2012 speedup on the other hand coincided with a 2 km retreat of the calving front (Figure 7B).

**5    Discussion**

Decomposition of the time-dependent velocity and surface elevation fields into distinct temporal scales reveals a repeating pattern on Jakobshavn where velocity and surface elevation variations originate at the terminus. The coincidence between speedup and slowdown of the glacier with thinning and thickening, respec-

430    tively, suggests a dynamic origin to the physical mechanism generating these variations. Prior studies have proposed that this mechanism is primarily characterized by a reduction of back stress at the terminus following a series of calving events, causing ice acceleration and increased driving stresses to propagate upstream which results in the observed high correlation between calving front position and velocity variations (e.g., Nick et al., 2009; Joughin et al., 2012; Bondzio et al., 2017). In this section, we detail the observed wave phenomena, introduce the proposition that the observed traveling waves are kinematic waves, and

435    discuss possible paths for future development of observational methods that will enable progression toward robust and efficient techniques for fusing remote sensing data from multiple sources and using *in situ* observations as prior information to constrain the inversions.

**5.1    Wave phenomena**

Our results indicate that velocity variations initiating at the terminus of Jakobshavn propagate upstream as traveling waves with

440    frequency-dependent propagation speeds (phase velocities) and attenuation length scales. To our knowledge, ours are the first results to explicitly quantify wave propagation at seasonal and multi-annual timescales using remote sensing observations, and, importantly, to show that traveling waves in this range of frequencies are dispersive, meaning that phase velocity is a function of frequency. These results on Jakobshavn complement our inferences of wave propagation for hourly to fortnightly timescale variations in the flow speeds of Rutford Ice Stream, Antarctica, using remote sensing data (Minchew et al., 2017), helping

445    to demonstrate the largely untapped potential of time-dependent remote sensing observations to quantify wave phenomena. Our ability to quantitatively observe wave propagation in glaciers using remotely sensed observations adds a new class of information and unique constraints on the mechanics of glacier flow — most notably the rheology of the ice-bed interface (*i.e.*, the form of the sliding law) and the rheology of natural glacier ice — for the simple reason that these mechanics influence both the state of any given glacier as well as the transient response of the glacier to external forcing. At the moment, data

450    sparsity  only allows for quantification of phase velocities and attenuation length scales for describing overall wave propagation behavior. As more data become available, the time series methods outlined above should allow for observations of waveforms manifest in surface elevation fields  constraints on dispersion relations (the relationship between frequency and wavelength)  on individual glaciers that cover a broader range of frequencies with finer resolution in the frequency domain. Realizing this

potential for remote sensing time series is important because the characteristics of wave propagation, specifically the dispersion relation as defined for a wide range of frequencies, are intrinsic properties of dynamical systems, if we define the system in this case such that it includes the glacier and boundary conditions. As such, time-dependent velocity and elevation data for glaciers characterized by a wide range of sliding speeds and geometries can be used to determine the relative contributions of forcing frequency, ice thickness, glacier width, and basal traction on measured phase velocities attenuation lengthscales, thereby providing a method for inferring relevant mechanical and rheological parameters.

[revised manuscript text omitted]

**5.2 Applicability to other study areas**

The GIMP velocity data over Jakobshavn Isbræ exhibits strong, well-defined short- and long-term variations, which facilitates reconstruction of the spatiotemporal evolution of the traveling waves discussed in this work. Additionally, the dense temporal sampling relative to the signals of interest avoids potential issues related to oversmoothing of short-term velocity variations. However, many other glaciers and ice streams in Greenland and Antarctica will not have the same level of data coverage as Jakobshavn, which may limit the recovery of similar dynamical signals. Data coverage in this context is specified by temporal sampling and spatial continuity of velocity data where the former is likely to be the primary limiting factor for time-series analysis. For example, velocity data derived primarily from optical platforms are generally restricted to the summer months where cloud and snow cover effects are minimized. This asymmetry in coverage for a given year will alias reconstruction of seasonal velocity cycles, which would likely cause artifacts when attempting to quantify wave properties like phase velocity. We estimate that velocity data provided at monthly intervals constitute the lower bound for temporal resolution in order to quantify wave behavior at sub-annual timescales using the methods presented here. Of course, higher phase velocities for certain classes of dynamical signals may necessitate remote sensing data with finer temporal resolution (Minchew et al., 2017)

Spatial resolution and spatial data gaps can also limit characterization of wave behavior and other changes in ice flow. For example, regions near glacier termini will undergo periods of missing data associated with termini retreat where velocity data cannot be obtained over open water. The temporal interpolation properties of B-splines can mitigate these effects to some degree, but study areas with more persistent spatial gaps will likely benefit from incorporation of spatial coherency, which enforces that neighboring grid points share similar temporal behavior. However, data that require stronger levels of spatial coherency may also result in reconstructed signals that are oversmoothed, which would bias phase velocities and decay lengthscales to lower and higher values, respectively. In these situations, it would be beneficial to incorporate independent data sources like GPS time-series to provide additional validation data for "tuning" the time-series analysis parameters. Overall, we expect that current and future remote sensing platforms will provide high-quality data similar to the GIMP data over Jakobshavn Isbræ, and we discuss those implications next.

**5.3 Future work in remote sensing**

The increasing availability of surface velocity and elevation fields sampled at monthly-to-sub-monthly timescales will continue to provide opportunities to study the rapid evolution of fast-flowing glaciers to various environmental forcings. The operational capabilities of several working groups that produce velocity fields over the Greenland and Antarctic Ice Sheets will consistently improve as new data are made available and techniques for generating velocity estimates are refined. In particular, the upcoming

NASA-ISRO Synthetic Aperture Radar (NISAR) mission will generate unprecedented volumes of data that are useful for quantifying surface change for a number of scientific applications, including glacier dynamics (NISAR, 2018). The wide imaging swath (~240 km) coupled with a 12-day repeat cycle and global coverage will allow for systematic observations of high-resolution velocity variations over interconnected glacier networks and coupled ice stream and ice shelf systems. Such observations will facilitate quantification of the spatiotemporal responses of glaciers and ice streams to any changes to the stress state, such as changes to the terminus position, loss of ice-shelf buttressing, changes in frictional properties of the bed, evolution of the subglacial hydrology. These processes will likely result in wave phenomena similar to those observed at Jabovshavn Isbræ (this study) and Rutford Ice Stream (Minchew et al., 2017) and would be well-observed with platforms like NISAR. Furthermore, the quantification of phase velocities and attenuation length scales at multiple forcing frequencies would provide valuable constraints on a general theory for wave propagation for fast-flowing glaciers because the characteristics of wave propagation are intrinsic properties of any given glacier system, which includes the boundary conditions.

The temporal resolution of surface DEMs is currently a limiting factor in quantifying sub-annual dynamical thinning. In this study, we noted that the thinning signal in the ice adjacent to the fast-flowing regions may be due to oversmoothing of the time series due to the limited temporal resolution of the ArcticDEM dataset. Therefore, elevation or altimetry datasets that have increased temporal sampling, such as IceSat-2, may help isolate the shorter-term dynamic signals from any longer-term SMB-based variations. In particular, future analysis would benefit from the 91-day repeat time of  ICESat-2 for capturing seasonal elevation variations for direct comparison and synthesis with the velocity seasonal variations. This type of analysis could lead to a full three-dimensional velocity time series, which has the potential to improve quantification of strain and stress fields, constraints on ice rheology, and assimilation of velocity data into state-of-the-art ice flow models. For glaciers and ice streams where a persistent ice shelf or tongue exists, tidal forcing may become an important stress perturbation, in which case accurate reconstruction of vertical displacements would be necessary in order to constrain the dominant tidal constituents of motion (Minchew et al., 2017).

The flexible time series framework described here introduces the potential for using *in situ* observations as prior information (encoded in the prior model covariance matrix $C_m$) in forming time-dependent surface velocity fields. One example of this synergy between remote sensing and *in situ* data is the use of GPS/GNSS observations to constrain the form of the temporal basis functions, as we did in Minchew et al. (2017). A similar constraint may be obtained from terrestrial radar instruments that record velocity variations at timescales of minutes, allowing for high-resolution observations of dynamic responses to calving events or mélange collapse (e.g. Xie et al., 2019; Cassotto et al., 2019). In those situations, temporal basis functions and spatial correlations between basis functions can be used for dictionary construction and time-series inversions. Another, less obvious, example is the potential for employing catalogs of calving events gleaned from seismic observations (Olsen and Nettles, 2017, 2019; Olinger et al., 2019) to constrain the timing and duration of transient accelerations in ice flow. Such constraints on the temporal evolution of the fields observed from remote sensing observations should afford novel opportunities to constrain phenomena such as the localization of strain rates (and, thereby, stresses) associated with fracture and calving. We expect the usefulness of the flexible methods we present here to grow as more remote sensing and *in situ* data become available.

**6  Conclusions**

We have presented a framework for forming continuous time-dependent surface velocity and elevation fields from publicly available surface velocity and elevation data. This framework is based on a sparsity-regularized linear regression method that reconstructs time series as a linear combination of relevant basis functions. The flexibility and expressive power of the basis function representation allows for accurate reconstruction of time series in the presence of noisy and missing data while also allowing for a natural decomposition of the total signal into signals of multiple temporal scales. Over Jakobshavn Isbræ, this decomposition permitted a detailed investigation into the spatiotemporal characteristics of the evolving seasonal cycle of ice speedup and slowdown  which are shown to be highly correlated to seasonal terminus variations. Analogously, longer-term changes in velocity were isolated and  also highly correlated with longer-term terminus variations. This type of analysis is directly applicable to many outlet glaciers in Greenland, Antarctica, and other areas where multitemporal remote sensing data is available and could improve our understanding of the dynamic response of glaciers to various geographic and environmental forcings.

We demonstrated that the time series reconstruction permitted the quantification of traveling wave propagation resulting from terminus forcing functions at different temporal frequencies. These results build upon an important new area of research that aims to achieve a mechanistic understanding of glacier flow from time-dependent velocity data. To our knowledge, our results are first to show from observations that waves on glaciers with seasonal to multi-annual periods are dispersive, with a ratio of observed phase velocities approximately equal to the square root of the ratio of frequencies. We hypothesize that the observed waves can be classified as kinematic waves based on their long periods (much longer than the viscoelastic relaxation time), correlation with changes in the terminus position, and coincident variations in surface velocity and elevation. However, the dispersive nature and higher phase velocities of the observed waves relative to previously proposed kinematic waves necessitates further investigation into their physical drivers and the overall dynamic response of glaciers to stress and mass perturbations. These observations of traveling waves are only possible due to the strong velocity response to changes in terminus position, as well as our ability to isolate short- and long-term signals in the velocity data. Looking forward, we aim to assimilate other velocity sources for Jakobshavn Isbræ (e.g., optical or Sentinel-1), as well as other elevation and altimetry data sets to improve temporal sampling and to obtain full 3D surface velocity time series. The resultant dataset will likely lead to a marked improvement in incorporating velocity data in ice flow models for simulation and inversion of mechanical properties.

*Code availability*. The time series analysis and decomposition tools are available at https://github.com/bryanvriel/iceutils.

*Data availability*. The velocity data are available at NSIDC (NSIDC-0481 at https://nsidc.org/data/measures/gimp, last access: Sep 2019). The original ArcticDEM elevation data are available from the Polar Geospatial Center, University of Minnesota (ArcticDEM at https://www.pgc.umn.edu/data/arcticdem, last access: Sep 2019) while the OMG DEM data are available at https://doi.org/10.5067/OMGEV-GLNA1 (last access: Oct 2019). The pre-2017 calving fronts obtained in this study are available from https://doi.org/10.1594/PANGAEA.897066

[revised manuscript text omitted]